# Antifungal Synergy: Mechanistic Insights into the R-1-R Peptide and *Bidens pilosa* Extract as Potent Therapeutics against *Candida* spp. through Proteomics

**DOI:** 10.3390/ijms25168938

**Published:** 2024-08-16

**Authors:** Yerly Vargas-Casanova, Claudia Patricia Bravo-Chaucanés, Samuel de la Cámara Fuentes, Raquel Martinez-Lopez, Lucía Monteoliva, Concha Gil, Zuly Jenny Rivera-Monroy, Geison Modesti Costa, Javier Eduardo García Castañeda, Claudia Marcela Parra-Giraldo

**Affiliations:** 1Microbiology Department, Faculty of Sciences, Pontificia Universidad Javeriana, Bogotá 110231, Colombia; y.vargasc@javeriana.edu.co (Y.V.-C.); claub06@gmail.com (C.P.B.-C.); 2Proteomics Unit, Universidad Complutense de Madrid, 28040 Madrid, Spain; sdelacam@ucm.es; 3Microbiology and Parasitology Department, Faculty of Pharmacy, Universidad Complutense de Madrid, 28040 Madrid, Spain; raquelml@ucm.es (R.M.-L.); luciamon@ucm.es (L.M.); conchagil@ucm.es (C.G.); 4Faculty of Sciences, Universidad Nacional of Colombia, Bogotá 111321, Colombia; zjriveram@unal.edu.co (Z.J.R.-M.); jaegarciaca@unal.edu.co (J.E.G.C.); 5Chemistry Department, Faculty of Sciences, Pontificia Universidad Javeriana, Bogotá 110231, Colombia; modesticosta.g@javeriana.edu.co

**Keywords:** bovine lactoferricin peptides, *Bidens pilosa*, *Candida albicans*, proteomics, ROS, disfunction mitochondrial, antifungal resistance

## Abstract

Previous reports have demonstrated that the peptide derived from LfcinB, R-1-R, exhibits anti-*Candida* activity, which is enhanced when combined with an extract from the *Bidens pilosa* plant. However, the mechanism of action remains unexplored. In this research, a proteomic study was carried out, followed by a bioinformatic analysis and biological assays in both the SC5314 strain and a fluconazole-resistant isolate of *Candida albicans* after incubation with R-1-R. The proteomic data revealed that treatment with R-1-R led to the up-regulation of most differentially expressed proteins compared to the controls in both strains. These proteins are primarily involved in membrane and cell wall biosynthesis, membrane transport, oxidative stress response, the mitochondrial respiratory chain, and DNA damage response. Additionally, proteomic analysis of the *C. albicans* parental strain SC5314 treated with R-1-R combined with an ethanolic extract of *B. pilosa* was performed. The differentially expressed proteins following this combined treatment were involved in similar functional processes as those treated with the R-1-R peptide alone but were mostly down-regulated (data are available through ProteomeXchange with identifier PXD053558). Biological assays validated the proteomic results, evidencing cell surface damage, reactive oxygen species generation, and decreased mitochondrial membrane potential. These findings provide insights into the complex antifungal mechanisms of the R-1-R peptide and its combination with the *B. pilosa* extract, potentially informing future studies on natural product derivatives.

## 1. Introduction

*Candida albicans* is a commensal yeast fungus that is a normal component of the human microbiota. It is commonly found on mucosal surfaces, including the oral cavity, gastrointestinal tract, respiratory tract, and genitourinary tract, as well as on the skin. When conditions disrupt the integrity of the mucocutaneous barriers or compromise the protective defense mechanisms of the host, such as through antibiotic-induced dysbiosis, iatrogenic immunosuppression, or other medical interventions, *C. albicans* might transition into an opportunistic pathogen. This can lead to a range of conditions, from superficial mucocutaneous diseases to severe, life-threatening systemic diseases [1,2]. 

*Candida* species are the leading cause of fungal infections in humans, with *C. albicans* being the most common pathogen associated with opportunistic infections [3]. Clinically important infections caused by *C. albicans* can be categorized as either mucosal or systemic. Major mucosal infections include vulvovaginal candidiasis (VVC), oropharyngeal candidiasis (OPC), esophageal candidiasis (EPC), and, less frequently, onychomycosis [1]. Mucosal candidiasis, especially VVC, can occur in immunocompetent people, although those who are immunocompromised are at a higher risk of more frequent, severe, or recurrent infections. It is estimated that VVC affects approximately 75 million women annually, with the recurrence rates ranging from 5 to 8% [2].

Systemic or invasive candidiasis (IC) impacts normally sterile body parts, such as the bloodstream (candidemia), and can involve the central nervous system (CNS), liver, spleen, heart, or kidneys. It may also affect the intra-abdominal compartment, with or without candidemia. Despite antifungal therapy [4], IC is associated with a high mortality rate. It is estimated that there are 1,565,000 cases annually, with at least 995,000 (63.6%) resulting in death [5,6].

Currently, the availability of antifungal agents is limited to polyenes, azoles, and echinocandins despite the significant impact fungi have on human health. This limitation is primarily due to the slow development of new antifungal agents. The slow pace is largely attributed to the complexity of fungal cells, which are eukaryotic, and the challenges associated with the permeability of compounds through the fungal cell wall and membrane [7].

Another issue with fungal medications is the increasing resistance observed in the fungus *Candida*. Fluconazole (FLC) is one of the most commonly prescribed antifungal drugs for *Candida* infections [8], and ~7% of all *Candida* blood samples tested at CDC are resistant to FLC [9,10]. The resistance of *C. albicans* to FLC can result from various factors, including mutations or the overexpression of the *ERG11* gene (the drug target), loss-of-function mutations in *ERG3* (which impedes the accumulation of toxic sterols), increased drug efflux due to up-regulation of ABC transporters, and aneuploidy, such as duplication of the left arm of chromosome 5 [8,11].

In the context of new drug discovery, combination antifungal therapy is regarded as a promising strategy. Drug combinations offer several advantages, including a reduced risk of developing antifungal resistance, enhanced antifungal activity at lower doses of the drug, shorter treatment duration, and decreased drug toxicity [12].

In a previous study [13], we assessed the antifungal efficacy of the palindromic peptide R-1-R: RWQWRWQWR, along with twenty synthetic derivatives and ethanolic extracts from the leaves and stems of *Bidens pilosa*. These were tested against clinical isolates of *C. albicans* and *C. auris*. Furthermore, we explored combinations of the peptide, extract, and/or FLC, and evaluated the cytotoxicity of the peptide and extract on erythrocytes and fibroblasts. Our results demonstrated that the original palindromic peptide, certain derived peptides, and the ethanolic extract of *B. pilosa* leaves exhibited significant activity against some of the tested strains. It is suggested that the antifungal activity of the peptide R-1-R may be attributed to its amphipathicity. For the *B. pilosa* extract, the observed activity might be due to a synergistic effect among the metabolites present, such as phenylpropanoids and flavonoids, which were previously identified through the chemical characterization of the extract.

On the other hand, the combination of peptide/FLC or peptide/extract/FLC successfully converted *C. albicans* strain 256 and *C. auris* strain 537 from an FLC-resistant to an FLC-sensitive phenotype. Additionally, combining the extract with the original palindromic peptide significantly reduced the minimal inhibitory concentrations (MICs) against *C. albicans* SC5314, *C. albicans* 256, *C. auris* 435, and *C. auris*. 537 by factors ranging from 2 to 16. These combinations also induced morphological changes in the cells, such as surface deformations. Our results suggest that the combinations of R-1-R/FLC, R-1-R/extract/FLC, and R-1-R/*B. pilosa* extract represent promising alternatives for enhancing the antifungal activity while reducing the cytotoxicity and costs. These strategies could potentially be effective in treating diseases caused by both sensitive and resistant *Candida* spp. [13].

The palindromic peptide R-1-R has demonstrated broad-spectrum activity, being effective against Gram-positive and Gram-negative bacteria, different species of *Candida*, and cancer cells from breast, colon, and cervical tissues [14,15,16,17,18,19]. This peptide shows considerable promise for the development of new treatments. However, its mechanism of action in fungi remains unexplored. The peptide R-1-R is a synthetic derivative of the minimal motif (RRWQWR) from Bovine Lactoferricin (LfcinB), which itself is derived from the hydrolysis of the N-terminal region of Bovine Lactoferrin (LFB) [20]. Therefore, it is plausible that the mode of action of the peptide R-1-R may resemble that of LfcinB and its related peptides. 

LfcinB has a net charge of +8 and amphipathic properties, enabling it to interact with plasma membranes through both electrostatic (Arginine and Lysine) and hydrophobic (Tryptophan) interactions [21]. Additionally, LfcinB has been shown to internalize into cells, causing alterations in ultrastructural features and promoting the aggregation of cytoplasmic materials [22]. This internalization has been linked to the stimulation of ATP synthesis and extracellular secretion, which ultimately leads to pore formation in the plasma membrane [22,23]. Peptides derived from LFB also exhibit antifungal activity primarily through interactions with the cell wall and plasma membrane. Scanning electron microscopy (SEM) revealed morphological changes on the surface of fungi, including perforations, the leakage of cytoplasmic content, membrane blebs, and deep pits [23]. Furthermore, confocal microscopy has demonstrated that these peptides can penetrate the surface of *C. albicans* [24]. Finally, Chang et al. [25] reported that LfcinB15, a peptide containing the minimal LfcinB motif, localizes to the cell surface and vacuoles within *C. albicans* after incubation. This peptide exerts its effects through various mechanisms, including cell membrane disruption, the induction of reactive oxygen species (ROS), and mitochondrial dysfunction. Additionally, two mitogen-activated protein kinases, Hog1 and Mkc1, were identified as being activated in *C. albicans* in response to LfcinB15 treatment.

Although the broad anti-*Candida* activity of *B. pilosa* extracts has been demonstrated, the mechanisms of action remain unclear. Angelini [26] conducted the only known research on this topic, where docking experiments revealed that caftaric acid, a metabolite present in a methanolic extract of *B. pilosa* leaves, exhibits a micromolar affinity for lanosterol 14α-demethylase, a key enzyme in fungal ergosterol biosynthesis.

Due to the limited information regarding the mechanisms of action of R-1-R, *B. pilosa* extract, and their combination, this study investigated the changes in protein expression in *C. albicans* SC5314 and *C. albicans* 256 (FLC resistant) using proteomic and biological approaches.

## 2. Results and Discussion

### 2.1. Susceptibility of C. albicans Mutant Strains to R-1-R and B. pilosa Extract

In previous studies, we reported the antifungal activity of the R-1-R peptide (MIC and minimal fungicidal concentration [MFC], 100 μg/mL) [17] and the extract of *B. pilosa* (MIC and MFC, 500 μg/mL) [13] against the reference strain *C. albicans* SC5314 and the fluconazole-resistant isolate *C. albicans* 256 (Table 1). These findings demonstrate that the resistance mechanism of the clinical isolate *C. albicans* does not interfere with the activity of the peptide.

According to other authors, LfcinB [20,22] and peptides derived from LfcinB [23,24,25,27] exhibit distinct mechanisms for killing *C. albicans*. These mechanisms include the disruption of the cell membrane, induction of ROS generation, and mitochondrial dysfunction. Consequently, mutants with alterations in these cellular processes were selected for treatment with the R-1-R peptide or *B. pilosa* extract (Table 1).

First, two efflux pump mutants (*cdr1∆/CDR1 and cdr2∆/CDR2*) were evaluated, and no changes were observed in the susceptibility to R-1-R (MIC/MFC: 100 μg/mL). However, a slight tolerance to the *B. pilosa* extract (MIC/MFC: 1000/2000- > 2000 μg/mL) was noted compared to the wild-type strain and the FLC-resistant isolate. FLC, used as a control and known to be a substrate for these efflux pumps, showed a significant increase in sensitivity (MIC: <0.25 μg/mL).

The results for FLC were as expected; however, they varied for the extracted peptide. In a previous study [13], a synergistic effect was observed with the R-1-R/FLC combination, indifference in the extract/FLC combination, and additivity in the R-1-R/extract/FLC combination. We hypothesized that the synergy occurred due to the increased efflux of FLC caused by the overexpression of membrane transporters, particularly Cdr1p and Cdr2p, which are associated with resistance to this antifungal. The peptide likely exerted an inhibitory effect on the efflux pumps, preventing the export of FLC from the yeast cells. In contrast, the extract might not have affected the efflux pumps, resulting in an indifferent effect. Thus, a change in the sensitivity of the mutants to the peptide, but not to the extract, was anticipated. To clarify these results, additional tests were conducted, as described below.

Consecutively, four mitogen-activated protein kinases (MAPKs) mutants *(hog1∆/HOG1*, *hog1∆/hog1∆*, *mkc1∆/MKC1*, and *mkc1∆/mkc1∆*) were evaluated, and, especially, the homozygous *hog1∆/hog1∆* and *mkc1∆/mkc1∆* were more sensitive to the peptide R-1-R exhibiting MIC/MFC at 15.5/25 μg/mL and 25/50 μg/mL, respectively. To the *B. pilosa*, the effect was opposite, since the four mutant strains were slightly more tolerant (MIC: 500–000/MFC: 2000- > 2000 μg/mL). The results of R-1-R against *hog1∆* mutants agree with the results obtained using other peptides such as HsT5, hBD-2, and hBD-3, where increased sensitivity was also seen, suggesting that Hog1p has a role in the cellular response to antimicrobial peptides (AMPs) [28,29]. However, they differ from the results obtained using the LfcinB15 peptide, where the *hog1∆* mutant was more tolerant, although they confirmed the participation of Hog1p by showing an increase in the phosphorylation of this protein when it was treated with the peptide [25]. Therefore, we can infer that it is possible that the two MAPKs also respond to the effect of the extract.

On the other hand, seven mutants lacking mitochondrial complex I and IV genes were evaluated (*ali1∆/ALI1*, *ali1∆/ali1∆*, *cox4∆/COX4*, *cox4∆/cox4∆*, *mci∆/mci∆*, *orf19.4758∆/orf19.4758∆*, and *orf19.7590∆/orf19.7590∆*), and the greatest changes in susceptibility were observed in *ali1∆/ALI1* and *cox4∆/COX4*, with greater sensitivity obtained to R-1-R (MIC/CMF: 12.5/25 μg/mL) and significant tolerance to the *B. pilosa* extract (MIC and MFC: >2000 μg/mL). The results suggest that in response to R-1-R and the extract, the mitochondrial electron transport chain (ETC) is being affected due to the change in the susceptibility of *ali1∆/ALI1* (related to mitochondrial complex I assembly) and *cox4∆/COX4* (putative cytochrome c oxidase) mutants [30]. Regarding the results obtained using R-1-R, similar to those observed with MAPKs, the Lfcinb15 peptide exhibited greater tolerance to the ETC mutants. However, it was found that this effect was associated with mitochondrial dysfunction [25].

The susceptibility of the homozygous mutant strains was also evaluated for the combination of the R-1-R peptide and *B. pilosa* extract, obtaining significantly greater inhibition of the *hog1∆* strain and less inhibition for the *mci∆* strain compared to SC5314 treated with the combination (Figure 1).

The haploinsufficient phenotype, or aptitude test (deletion of an allele in a diploid to obtain a mutant strain), is a tool that has been widely studied in *S. cerevisiae* and represents an alternative method to deepen the understanding of *C. albicans*. It has been observed that when a new inhibitory compound is tested, the response (hypersensitivity and resistance to the compound, i.e., haploinsufficiency and haplocompetence, respectively) of specific heterozygous strains can provide phenotypic information that reflects the mechanism of action of the compound [31].

Based on the findings, the results obtained using the mutant strains suggest that the R-1-R peptide and the *B. pilosa* extract exhibit a mechanism of action related to cell wall stress mediated by MAPKs (Hog1p and Mkc1p) [32] and alteration in mitochondrial respiration due to interruption of the ETC. 

Additionally, it is suggested that the mechanism of action for the combination may be related to cellular stress, regulated by Hog1 MAPK, and similar to the effect of the peptide and extract alone, where an alteration of the ETC could be occurring.

To investigate the possible mechanism of action of R-1-R and its combination with *B. pilosa* extract against *C. albicans* in more detail, changes in the protein expression profiles of both FLC-sensitive and -resistant *C. albicans* were analyzed using label-free proteomics after six hours of incubation with R-1-R or the combination. 

**Table 1 ijms-25-08938-t001:** MICs and MFCs of R-1-R and extract of *B. pilosa* against *C. albicans* mutants.

**Systematic Name**	**Description ^1^**	**Strain**	**R-1-R**	***B. pilosa* Extract**	**FLC**
			MIC/MFC (μg/mL)		
	Wild type	*C. albicans* ATCC SC5314 ^2^	100/100	500/500	1
	Oral clinical isolate, FLC resistant	*C. albicans* 256 PUJ-HUSI ^2^	100/100	500/500	64
orf19.6000	Multidrug transporter of ABC superfamily; transports phospholipids in an in-to-out direction	*cdr1∆/CDR1*	100/100	1000/2000	<0.25
orf19.5958	Multidrug transporter, ATP-binding cassette (ABC) superfamily; transports phospholipids in an in-to-out direction; overexpressed in azole-resistant isolates	*cdr2∆/CDR2*	100/100	1000/>2000	<0.25
orf19.895	MAP kinase of osmotic-, heavy metal-, and core stress responses; role in regulation of glycerol and D-arabitol in response to stress	*hog1∆/HOG1*	50/100	1000/2000	4
		*hog1∆/hog1∆*	12.5/25	500/2000	ND
orf19.7523	MAP kinase, role in cell wall structure/maintenance and caspofungin response; phosphorylated on surface contact, membrane perturbation, or cell wall stress	*mkc1∆/MKC1*	50/100	1000/2000	2
		*mkc1∆/mkc1∆*	25/50	1000/>2000	ND
orf19.1710	Putative NADH-ubiquinone oxidoreductase; plasma membrane localized; protein decreases in stationary phase	*ali1∆/ALI1*	12.5/25	>2000/>2000	<0.5
		*ali1∆/ali1∆*	50/100	500/500	ND
orf19.1471	Putative cytochrome c oxidase subunit IV; Mig1 regulated	*cox4∆/COX4*	12.5/25	>2000/>2000	<0.5
		*cox4∆/cox4∆*	50/200	500/2000	ND
orf19.2570	Putative NADH-ubiquinone dehydrogenase	*mci∆/mci∆*	100/200	500/2000	ND
orf19.4758	Putative reductase or dehydrogenase	*orf19.4758∆/orf19.4758∆*	100/200	500/1000	ND
orf19.7590	Putative NADH-ubiquinone oxidoreductase; identified in detergent-resistant membrane fraction (possible lipid raft component)	*orf19.7590∆/orf19.7590∆*	100/100	500/500	ND

^1^ The information of the mutant strains was obtained from [30]. ^2^ MIC and MFC values were reported in the reference [13]. ND: not determined.

### 2.2. Identification of C. albicans Total Proteins

The reference strain, *C. albicans* SC5314 (antifungal sensitive), was cultured under three conditions: (1) a basal condition without treatment, (2) in the presence of the peptide R-1-R at a concentration of 100 µg/mL, corresponding to the minimum inhibitory concentration (MIC), and (3) in the presence of R-1-R in combination with *B. pilosa* extract at concentrations of 25 µg/mL and 250 µg/mL, at which a synergistic effect had been previously observed.

The clinical isolate *C. albicans* 256 (FLC resistant) was cultured under two conditions: basal (without treatment) and with the peptide R-1-R at a concentration equal to the MIC of 100 µg/mL. The MIC values and concentrations demonstrating synergistic or additive effects in the two *C. albicans* strains were previously reported by our group [13].

Four replicates of total cytoplasmic protein extracts from both strains under these conditions were obtained and quantitatively analyzed using the Bradford method and SDS-PAGE. The results indicated consistent protein concentrations for all study samples (Appendix A). Furthermore, the bands were clear and uniform, the proteins were not degraded, and the parallelism of each line was good (Appendix A). This indicates that the total protein of *C. albicans*, for both SC5314 and 256, met the quality standard of the experiment. The difference in total protein obtained between the SC5314 and the resistant strain, as well as between the untreated and treated strains, aligns with a previous study [33] in which the Lowry method was used to estimate the total protein levels for *C. albicans* isolates resistant to antifungals, and found that strains resistant to FLC and itraconazole had a higher protein concentration than those resistant to ketoconazole or Amphotericin B. Additionally, when the yeasts were incubated with the antifungal, there was a decrease in the total protein concentration compared to untreated yeasts. Both studies indicate that differences in the total protein concentration are related to drug resistance, which, according to Zaidi et al. [33], could be used as a marker for the development of resistance.

### 2.3. Proteomic Analysis of C. albicans SC5314 and 256 after Treatment with R-1-R

A comparison of SC5314 and strain 256 treated with R-1-R revealed that the 256 resistant strain exhibited a greater number of exclusive proteins both under basal conditions (115 vs. 75 proteins) and after R-1-R treatment (331 vs. 238 proteins) (Figure 2a,b). In all comparisons, proteins considered exclusive and significant had a q value < 0.05. Differential quantitative analysis of the proteins regulated using R-1-R treatment identified 245 up-regulated and 88 down-regulated proteins in SC5314 (Appendix A). In strain 256, 335 up-regulated and 122 down-regulated proteins were observed (Appendix A). This comparison (Figure 2c) highlights several key points: (i) Both strains show a higher number of up-regulated proteins compared to down-regulated proteins, and the fact that more up-regulated proteins were found when the strains were incubated with R-1-R suggests that the yeast could be generating a compensatory response to the damage caused by the peptide. (ii) The most abundant regulated proteins in both strains are those related to metabolism and nuclear functions, comprising 21–40% of the total. (iii) In SC5314, the most prevalent up-regulated proteins are cell wall or membrane proteins (16%), membrane transporters (7%), oxidative stress-related proteins (5%), and potential mitochondrial membrane proteins (3%), with similar trends observed in strain 256 but at lower percentages. (iv) Additionally, down-regulated proteins differed between strains. SC5314 showed greater down-regulation in membrane transport proteins (13%), followed by structural membrane or cell wall proteins (9%) and oxidative stress proteins (4%), while mitochondrial membrane potential proteins were unaffected. Conversely, strain 256 displayed more pronounced down-regulation in structural membrane or cell wall proteins (14%), followed by oxidative stress proteins (11%), membrane transporters (7%), and mitochondrial membrane potential proteins (2%).

After confirming that R-1-R predominantly impacts critical cellular processes at the cell surface, within the nucleus, and in the mitochondria across both strains, a functional annotation enrichment analysis was conducted for all regulated proteins (both up-regulated and down-regulated) under each condition (see Appendix A). The enriched Gene Ontology (GO) categories, including cellular component, molecular function, and biological process, are presented in Appendix A and illustrated in Figure 3.

For SC5314 treated with the peptide, the up-regulated proteins (Figure 3a) were associated with 45 biological process (PB) terms, 3 molecular function (MF) terms, and 12 cellular component (CC) terms. Notably, these terms highlighted lipid biosynthetic processes, cell wall biogenesis or organization, and the biosynthesis and metabolism of ergosterol, steroids, and phytosteroids, as well as external structure/encapsulation organization (indicated by green vertical bars). In contrast, the down-regulated proteins (Figure 3b) were associated with eight PB terms, three MF terms, and one CC term, with a focus on the biosynthesis and metabolism of long-chain fatty acids.

On the other hand, for the 256 proteins treated with R-1-R up, the analysis (Figure 3c) revealed the enrichment in 110 terms related to biological processes (BPs), 21 related to molecular functions (MFs), and 27 related to cellular components (CCs). This enrichment highlights the cellular and intracellular anatomical entities, gene expression, metabolic processes of nucleic acids, and ribosome biogenesis. Conversely, the down-regulated proteins (Figure 3d) showed enrichment in 12 terms related to BP, with no significant enrichment in the MF or CC terms, primarily emphasizing iron ion transport.

Subsequently, an in silico protein–protein interaction analysis was performed with the up- and down-regulated proteins in SC5314 and 256 treated with R-1-R (Appendix A). For SC5314 treated with R-1-R (Figure 4a), 96 proteins were involved, and 27 clusters were found. Of these, four clusters corresponded to membrane-related pathways, including glycerophospholipid metabolism (Cds1p, C2_00790cp_a, and Slc1p), down-regulation and steroid biosynthesis (Erg11p, Erg2p, Erg26p, Erg27p, and Erg5p), and biosynthesis and the metabolism of fatty acids (Cem1p, Fas1p, and Fas2p), which were up-regulated. On the other hand, one cluster was involved in the pathways of the respiratory chain and internal mitochondrial membrane complex, whose proteins C1_05180cp_a, C1_08080cp_a, Cox1p, C2_09510cp_a, and Mas1p were up-regulated, while C1_09980cp_a was down-regulated. Eight clusters were related to nucleic acid processes, especially RNA, and here, two up-regulated proteins stand out (Rfa1p and Hys2p) for participating in nucleotide excision repair, and the cluster formed by C4_02090cp_a (down-regulated), C1_00330cp_a, and Not4p (up-regulated) is involved in RNA degradation. The remaining clusters refer to other metabolic processes.

In the case of 256 (Figure 4b), 140 proteins were involved, and 42 clusters were found. A total of 4 clusters were correlated with the cell membrane, including pathways of glycerophospholipid metabolism (Cds1p, C2_00790cp_a, Sct1p, and Slc1p), phosphatidylinositol signaling (C2_02000wp_a, Sac1p, and Stt4p), steroids (Erg27p and Erg5p) and sphingolipid metabolism (Lag1p and Sur2p), with all up-regulated, and 17 cluster (71) proteins were related to nucleic acids, highlighting processes such as nucleotide repair (Ddc1p, Rfc1p, Rfc2p, and Rfc5p), DNA repair (Eaf7p, Msb2p, and Swc4p), and the degradation of RNA (Ccr4p and C1_00330cp_a). Interestingly, two clusters associated with autophagy (Lcb2p and Tor1p) and endocytosis (Gea2p and Sec7p) were also found, with both being up-regulated. Finally, 14 clusters comprising 39 proteins were related to other metabolic processes.

### 2.4. Important Proteins Regulated in SC5314 and 256 in Response to R-1-R

Table 2 summarizes the key differentially regulated proteins involved in the biological processes affected by the peptide R-1-R in both SC5314 and strain 256. In the context of cell wall processes, most proteins were up-regulated in both strains SC5314 and 256. These proteins are associated with cell wall biogenesis, regeneration, response to damage, organization, assembly, and maintenance, as well as adhesion, biofilm formation, and B-glucan biosynthesis. Biosynthesis of mannoproteins was found exclusively in SC5314 or 256, and only one protein related to chitin remodeling was found in 256.

To the membrane cell, after treatment with R-1-R, we found that the primary affected pathway was ergosterol biosynthesis in both FLC-sensitive and FLC-resistant strains. All proteins within this pathway were up-regulated. Notably, in strain 256, Erg11p was absent when treated with the R-1-R peptide, possibly due to a mutation in *ERG11* that confers FLC resistance to this clinical isolate. Fatty acid biosynthesis was also affected in both study strains, with both down-regulated and up-regulated proteins. 

Transport proteins, including those involved in the activation of efflux pumps such as Cdr1, carbohydrate transport proteins, and iron and cation transport proteins, were down-regulated in both strains. Notably, proteins involved in membrane transport were differentially regulated in SC5314 and 256. Among these, proteins related to ABC transporters were particularly highlighted, as they play a role in resistance to fluconazole (FLC).

In response to oxidative stress, both study strains exhibited an up-regulation of proteins associated with the Hog1 and Cap1 pathways and antioxidant activity. Conversely, proteins involved in oxidoreductase activity were down-regulated. Other differentially regulated proteins, related to similar processes, were found exclusively in either SC5314 or 256. A notable finding occurred when evaluating mitochondrial membrane potential: only two proteins involved in the respiratory chain and Cytochrome C activity were found in both strains, while other proteins were unique to each strain. In SC5314, all proteins associated with the respiratory chain were up-regulated. In strain 256, we observed up-regulation of Cytochrome C and proteins from the ETC complexes III and IV. Additionally, both SC5314 and 256 strains evidenced the up-regulation of proteins related to mitochondrial autophagy, mitophagy, autophagy, and the maintenance of mitochondria and vacuoles.

Finally, proteins associated with the nucleus or nucleic acid-related cellular processes were predominantly up-regulated in both study strains. Notably, this up-regulation included proteins involved in ribosomal biogenesis proteins, nuclear export, DNA repair, and RNA degradation. Regarding the down-regulated proteins, they were primarily involved in DNA replication, ribosome subunit formation, and an MAPK transcription factor related to the Hog1 signaling pathway. Two proteins exclusively up-regulated in SC5314 were identified, both of which play roles in repairing DNA damage caused by oxidation. In strain 256, the up-regulated proteins were involved in RNA degradation, replication, and ribosomal biogenesis, while the down-regulated proteins were associated with tRNA biosynthesis, nucleocytoplasmic transport, and ribosome biogenesis.

### 2.5. Effect of the Combination between R-1-R and B. pilosa Extract against C. albicans SC5314

After analyzing the potential effects of the R-1-R peptide on *C. albicans* strains that are both sensitive and resistant to FLC, we explored the mechanism of action of the combination of this peptide with an extract of *B. pilosa*. This combination has been shown to enhance anti-*Candida* activity.

*C. albicans* SC5314 was then treated with a combination treatment, which resulted in the identification of 118 proteins exclusive to the untreated strain and 115 proteins exclusive to the combination treatment (Figure 5a). This number represents approximately half of the proteins identified when the strain was treated solely with the R-1-R peptide (238 proteins) (Figure 2a). Differential quantitative analysis revealed 138 up-regulated proteins and 158 down-regulated proteins (Appendix A). Notably, the regulation of proteins observed was opposite to the results obtained using the peptide R-1-R, where more proteins were up-regulated compared to down-regulated (245/88 proteins, respectively). This suggests that the combination treatment suppresses a greater number of proteins, potentially indicating a reduced ability of the strain SC5314 to mount a compensatory response.

Subsequent functional categorization of the proteins revealed that among the up-regulated proteins (Figure 5b), their distribution was similar to that observed with R-1-R treatment (Figure 2c). The highest up-regulation was found in metabolism- and nucleus-related proteins, accounting for 38% and 28%, respectively. This was followed by membrane/cell wall proteins at 16%, mitochondria/oxidative stress and membrane transporters at 7% each, and mitochondrial membrane potential proteins accounting for 1%. In contrast, the down-regulated proteins exhibited a marked difference compared to R-1-R treatment. Specifically, a larger proportion of down-regulated proteins were associated with other metabolic processes (48%), which suggests a need for further investigation in future studies. Additionally, membrane potential proteins were affected (3%), a phenomenon not observed with peptide treatment alone (0%).

Regarding the GO analysis of SC5314 treated with the combination, up-regulated proteins (Figure 6a) were enriched in 14 terms associated with PBs and 7 with cellular components (CCs). Notably, the enriched terms included organization processes of the external encapsulating structure and cell wall organization. In contrast, treatment with the peptide affected both the cell wall and cell membrane processes but not the external region (Figure 3a).

For down-regulated proteins (Figure 6b), 24 terms associated with BPs, 6 MFs, and 3 CCs were enriched, highlighting the oxidoreductase activity and the fatty acid synthase complex, which are crucial in fatty acid biosynthesis. Similar terms were observed with R-1-R treatment, specifically those related to fatty acids (Figure 3b).

Finally, in the in silico analysis of protein–protein interactions in SC5314 treated with the combination, (Figure 6c), 97 proteins were involved, and 27 clusters were found. Three clusters were correlated with the cell membrane, including processes such as ergosterol biosynthesis [Erg1p (down), Erg2p, Erg26p, and Erg5p (up)], fatty acid beta oxidation [Ald5p and Hpd1p (down)], and fatty acid biosynthesis [Fas1p and Fas2p (down)]. One cluster (Gam1p and Xog1p (up)) was related to the cell wall. One cluster (C5_04530wp_a and Sod1p) was related to oxidative stress, where the superoxide dismutase pathway was down-regulated, indicating reduced antioxidant activity. Furthermore, a cluster was related to complex I of the mitochondrial respiratory chain, where Cr_01300wp_a, C2_00700wp_a, C1_09980cp_a, C6_02740wp_a, Nuo1p, and Yme1p were up-regulated, and only C1_08080cp_a was down-regulated. Also, 6 clusters contained 44 proteins involved in processes with nucleic acids, and 14 clusters included 33 proteins that are related to other metabolic processes. For this analysis, we highlighted the cell wall and oxidative stress clusters that were not seen in the protein interactions when the same strain was treated with the R-1-R peptide.

### 2.6. Important Proteins Up- or Down-Regulated for SC5314 in Response to the Combination

Important proteins that were exclusively regulated by the combination treatment and absent when SC5314 was treated with R-1-R alone were selected for further analysis. A smaller subset of proteins was chosen (Table 3).

At the cell wall level, the treatment of SC5314 with the combination led to the up-regulation of GPI anchor and cytoskeleton proteins. The down-regulated proteins included various cytoskeletal proteins, a chitinase, and a cell wall integrity protein. In the cell membrane, down-regulated proteins comprised one involved in ergosterol biosynthesis and a sterol transfer protein. Regarding membrane transporters, one protein associated with cation transport was up-regulated, whereas those down-regulated were linked to carbohydrate transport and MDR1 transporters.

Concerning oxidative stress response proteins, only two proteins were up-regulated, with one being associated with the HOG1 pathway. The down-regulated proteins included those involved in oxidative stress, oxidoreductases, and antioxidant activity. An up-regulated protein from complex III of the ETC was identified, while down-regulated proteins included those from complexes I and II of the ETC. Another protein was also associated with the ETC, and the last one was related to ATP synthesis.

In processes related to nucleic acids, all proteins were down-regulated, including those involved in ribosome function, rRNA maturation, and nucleocytoplasmic transport.

Overall, although similar processes or cellular organelles were affected by the R-1-R peptide and its combination with the extract, the latter notably resulted in a higher proportion of down-regulated proteins. Additionally, the combination treatment was characterized by a predominantly down-regulated expression of proteins. This observation underscores the significant role of the *B. pilosa* extract in enhancing the peptide’s anti-fungal efficacy, leading to a more pronounced lethal effect.

The results of this proteomic study preliminarily indicate that similar to LF and Lfcin or other LfcinB peptides [25,34], the R-1-R peptide compromises the cell wall and membrane, leading to oxidative stress and alterations in the mitochondrial membrane potential. Additionally, it was observed that R-1-R impacts the function of membrane transporters. Moreover, the observed effects on different functional protein groups were consistent regardless of whether the strain was sensitive or resistant to FLC with the exception of mitochondrial membrane potential damage.

To further explore the effects on the cell wall and membrane, efflux pumps, oxidative stress, and mitochondrial respiration, complementary studies were conducted using results from mutant strains and proteomic approaches.

### 2.7. Scanning Transmission Electron Microscopy (STEM)

STEM imaging revealed significant ultrastructural changes in *C. albicans* SC5314 treated for 2 h with the R-1-R peptide (Figure 7b,c) compared to the untreated control (Figure 6a). These changes included the presence of microbodies in the cytoplasm and a noticeable thickening of the cell wall. Treatment with *B. pilosa* extract also resulted in the presence of cytoplasmic microbodies and irregularities in the cell wall (Figure 7d) along with nuclear swelling and irregularities without microbodies (Figure 7e). The combination treatment led to swollen nuclei, which indicate potential cell bursting and the expulsion of cytoplasmic contents accompanied by cell wall rupture (Figure 7f–i), demonstrating a synergistic effect. R-1-R was used at 25 µg/mL, while *B. pilosa* extract was administered at 250 µg/mL.

The observation of *C. albicans* 256 using STEM showed that the untreated strain (Figure 8a,b) had a thicker cell wall compared to *C. albicans* SC5314 (Figure 7a). This difference may be influenced by the FLC resistance exhibited by this strain. In contrast to *C. albicans* SC5314, treatment with R-1-R (Figure 8c,d), the extract (Figure 8e,f), and the combination treatment (Figure 8g,h) resulted in a disorganized cytoplasm with a granular appearance surrounded by filaments, along with retracted and irregular cell membranes and walls. These alterations were more pronounced with the combination.

The irregularities observed in the cell walls of *C. albicans* SC5314 and *C. albicans* 256 following all three treatments align with findings from a previous study [13], where SEM revealed cell membrane retraction and wall alterations in the same strains when treated with R-1-R, an extract, and their combination.

This study is the first to observe the effects induced by the peptide, extract, and their combination on yeast cells. Consistent with previous studies on other antifungal molecules, the presence of microbodies indicates apoptosis-related cell death [35]. Additionally, the disorganized cytoplasm with a granular appearance suggests necrosis [36].

### 2.8. Alteration of the Activity of Efflux Pumps

The accumulation of Rhodamine 6G (R6G) in growing *C. albicans* cells is inversely correlated with the expression level of the ABC transporter *Candida* drug resistance 1 (CDR1) mRNA. The measurement of released R6G is used to determine the activity of these R6G-dependent ATP pumps [37].

Figure 9 shows that the treatment of *C. albicans* SC5314 with R-1-R (sub-MIC 25 µg/mL) significantly reduced R6G release. In contrast, treatment with *B. pilosa* extract (250 µg/mL) increased its release. This suggests that the peptide may inhibit efflux pump activity, whereas the *B. pilosa* extract appears to enhance it. Similar effects were observed for the FLC-resistant strain *C. albicans* 256 (Figure 9b). Controls without glucose exhibited no R6G release.

The efflux transporter Cdr1, activated by ATP and proton flow, is overexpressed and associated with FLC resistance. Previous studies have evaluated the antifungal activity of FLC combined with LF, Lfcin, or related peptides, demonstrating synergistic effects [34,38,39]. It has been suggested that LF and its derivatives can inhibit these transporters by dissipating the proton gradient across the cell membrane and inhibiting glucose uptake. However, further investigations are warranted. The results of this study confirm the ability of peptides derived from LfcinB to inhibit efflux pumps in *C. albicans*.

Additionally, we hypothesize that the extract enhances multidrug efflux, reducing azole intracellular accumulation and weakening the azole antifungal activity. Given that glucose metabolism produces ATP, higher basal glucose levels (observed at 0 min) may result in increased ATP production, thereby enhancing the activity of efflux pumps. Also, these glucose levels could have influenced the expression of genes involved in the regulation and synthesis of efflux pumps [40]. Consequently, the extract increased the efflux pump activity and affected the efficacy of compounds that are substrates of these pumps. Therefore, the *B. pilosa* extract likely does not affect the efflux activity that is mediated by ABC transporters likely due to the overexpression of *C. albicans* efflux pumps, which facilitates the efflux of FLC to the extracellular space and inhibits its intracellular accumulation [41]. Conversely, the peptide may inhibit the efflux pump activity, and the specific inhibition of ABC efflux pumps is a possible way to block multiple-drug resistance.

### 2.9. Induction of Cellular ROS Generation

Increasing evidence shows that several fungal species augment the production of ROS upon contact with antifungal drugs, including azoles, polyenes, and echinocandins [42] as well as AMPs [43,44,45] and peptides derived from LfcinB [25]. Multiple studies report that mitochondria are the origin of ROS in these cases, enhancing the fungicidal capacity of the drugs and contributing to drug-induced programmed cell death (PCD). However, it remains unclear whether ROS production is a primary mechanism of antifungal action or a secondary consequence of PCD. Additionally, ROS induction may promote tolerance to antifungal drugs as part of adaptive evolution [42].

In the current study, intracellular ROS accumulation was measured after 2 h of treatment (Figure 10a). Firstly, it is evident that both the positive control [Amphotericin B (AmB)] and the R-1-R peptide generated an increase in ROS production in SC5314; the increase was dependent on the concentration. These results are consistent with those obtained using the peptide Lfcinb15 [25]. In strain 256, the increase in ROS was significantly more pronounced, even at sub-inhibitory concentrations. This marked accumulation of ROS in strain 256 has been previously reported, suggesting that although the absence of ergosterol in this strain is not specifically related to FLC resistance, mutations in ergosterol biosynthesis genes may lead to an abnormal membrane structure and functionality. Consequently, these mutations could affect tolerance to oxidative stress [46].

In the case of the extract (Figure 10a), no significant increase in ROS was observed. However, based on the results from Table 1 with the mutant strains, it is evident that oxidative stress is being generated, which may be masked by the polyphenolic effects of the *B. pilosa* extract, known for its antioxidant activity [47]. When the reference strain and the FLC-resistant strain were treated with the combination, an increase in ROS was observed only in strain 256. This suggests that in SC5314, the extract could be exhibiting an antioxidant effect against the ROS generated by the R-1-R peptide. However, in strain 256, the antioxidant effect of the extract is insufficient, as the ROS generated by the peptide exceed the level observed in the FLC-susceptible strain.

Acknowledging the role of Hog1p and Mkc1p and the results obtained using the Lfcib15 peptide in response to oxidative stress, two heterozygous mutant strains were exposed to the peptide, extract, and combination (Figure 10a). The most significant results were observed with the R-1-R peptide at 100 µg/mL, where the accumulation of ROS decreased significantly in both mutant strains. H_2_O_2_ is a known ROS inducer; under its action, Hog1 phosphorylates Mkc1. However, cells lacking Mkc1 are not sensitive to oxidative stress [48], suggesting, in this study, that the sensitivity observed in the Mkc1 mutant strain in the presence of the R-1-R peptide is more related to oxidative stress than to cellular wall. Although Hog1 correlates with oxidative stress, it has been found not to be specific to this pathway. These findings could possibly explain why the mutant strains of these two kinases generate an amount of ROS similar to that of the WT strain, indicating that they are competent in responding to oxidative stress, possibly through the activation of the Cap1 pathway [49].

ROS in high quantities lead to oxidative stress, damaging vital components of cells and causing cell death [49,50]. Subsequently, a ROS-scavenging agent, N-acetylcysteine (NAC; 60 mM), was used to confirm that the fluorescence increased due to ROS and to correlate the antifungal activity of R-1-R with ROS. When NAC was added to the two strains pretreated with R-1-R and the combination, no increase in fluorescence was evident, confirming the relationship between fluorescence and ROS. Conversely, when the strains were treated with R-1-R and NAC (Figure 10b), an increase in cell viability was exhibited for SC5314, similar to the LfcinB15 peptide [25]. 

These results suggest that the ROS induced by R-1-R are directly related to the candidacidal activity of the peptide. While the initial mechanism involves the cell membrane, as previously stated, it is hypothesized that as an effector mechanism, the generation of ROS damages vital cellular components in yeast, promoting cell death. This phenomenon has been documented for antifungals such as itraconazole against *C. albicans* and Amphotericin B against *Aspergillus fumigatus* [43].

Moreover, after pretreatment with NAC, strain 256 remained susceptible to the peptide, confirming that the FLC-resistant strain has a disadvantage in regard to its tolerance to the peptide compared with the sensitive strain. This difference is possibly due to the mutation present in the ergosterol pathway, which induces mitochondrial dysfunction, as confirmed in the subsequent experiment.

### 2.10. Decreased Mitochondrial Membrane Potential

It is believed that the main source of ROS in fungal cells, following the action of antifungals, is mitochondria, when superoxide is formed after the leakage of electrons from respiratory complexes I and III in the ETC, where most of this superoxide is converted to hydrogen peroxide thanks to mitochondrial superoxide dismutases (SODs). Superoxide and hydrogen peroxide can remain within the mitochondria or diffuse into the cytosol, where they are neutralized by cytosolic SODs and catalases, respectively [42]. The excessive production of ROS causes alterations in the ETC, mitochondrial membrane potential (ΔΨm), and ATP generation, leading to mitochondrial dysfunction [49].

Given that the R-1-R peptide induces ROS accumulation, and it is presumed that the extract and their combination do so as well, we measured the ΔΨm of *C. albicans* SC5314, 256, and mutant strains following exposure to these treatments. This was achieved using the fluorescent dye rhodamine 123 (Rho 123), where an increase in the fluorescence intensity indicates mitochondrial membrane depolarization.

Several observations were made (Figure 11): Firstly, for the group of untreated strains, it was observed that compared to SC5314, the FLC-resistant strain and the two mutant strains exhibited an increase in the MFI; that is, at the baseline, they already present a low mitochondrial dysfunction; for the 256 strain, it is possible that this effect is related to the alteration in the membrane due to resistance to FLC, and for the *Ali1* and *Cox4* mutant strains, this occurs because they are ETC-deficient mutants. This finding corresponds with the proteomic analysis, suggesting that in the resistant strain, a greater percentage of down-regulated proteins related to mitochondrial membrane potential were detected. Conversely, the WT strain treated with the peptide exhibited no down-regulated proteins.

Subsequently, it was observed that for SC5314, as expected, the positive control, sodium azide, increased the fluorescence. With the R-1-R peptide, the increase in the MFI was modest, while with the extract, a significant increase was observed depending on the concentration, and with the combination, the MFI was significantly higher, even exceeding that of the positive control and the extract alone. This indicates that although mitochondrial dysfunction occurs in all cases, this effect is more pronounced with the extract and the combination. These results are consistent with the increased MFI observed for a peptide derived from LfcinB [25]. For strain 256, regardless of the treatment concentration evaluated, the fluorescence was significantly higher than for SC5314, indicating that this strain indeed shows greater susceptibility to mitochondrial dysfunction.

To further investigate the functionality of mitochondria in this study, two heterozygous mutant strains (*ali1Δ/ALI1* and *cox4Δ/COX4*) from the ETC complexes I and IV were evaluated. In each case, an increase in the MFI was noted compared to the WT. As shown in Figure 10, the amount of Rho 123 incorporated and the fluorescence intensity increased in both mutants when treated with the *B. pilosa* extract, indicating compromised mitochondrial membrane integrity compared to the mutants treated with R-1-R, which exhibited a lower mitochondrial membrane potential. These findings are consistent with a previous investigation within our group involving the same mutant strains treated with varying concentrations of piperine isolated from a natural extract [45]. Notably, the dynamics of Rho 123 accumulation in the mitochondria of *ali1Δ/ALI1* and *cox4Δ/COX4* cells differed from those treated with the combination, which exhibited the highest mitochondrial membrane potential. Taken together, these results suggest that both compounds induce the increased depolarization of the mitochondrial membrane, where *Ali1* and *Cox4* may play a significant role in the action of *B. pilosa*, whereas susceptibility to R-1-R in fungi may be determined through different mechanisms.

## 3. General Discussion 

It has been described that the main antifungal mechanism of AMPs targets the cell surface, affecting the cell wall or membrane as well as intracellular targets such as nucleic acids and proteins [37,38]. For LfcinB peptides, the anti-*Candida* activity is attributed to cell surface damage, which induces the generation of reactive oxygen species (ROS) and mitochondrial dysfunction [25].

Previously, we found that the R-1-R peptide caused the disruption of the cell surface with permeabilization. In contrast, the *B. pilosa* extract did not exert this effect. However, when the combination was evaluated, permeabilization of the cell surface was observed in both antifungal-sensitive and FLC-resistant *C. albicans* [13]. Considering these results and the proposed general mechanism of action for LfcinB, which involves electrostatic and hydrophobic interactions with cell membranes [20], our hypothesis is that the R-1-R peptide binds to the cell surface, causing permeabilization and initiating candidacidal activity. This permeabilization allows for the entry of other molecules such as FLC [51] or *B. pilosa* extract into the yeast, where they act in the effector phase of the antifungal activity.

Additionally, the proteomic analysis revealed that important processes of both the cell wall and membrane are affected by the R-1-R peptide (Figure 12). For example, the biosynthesis of mannoproteins, glucans, chitin, GPI-anchored proteins, ergosterol, and other fatty acids were affected independent of the antifungal resistance phenotype. Likewise, it is evident how most of these damages are preserved when treated with the combination and where it is additionally seen that it is possible that the extract intensifies the damage by also affecting the integrity of the cytoskeleton.

It is important to emphasize that the cell wall is a dynamic structure with significant plasticity, which allows various cell morphologies, molecular remodeling, and changes in the composition of the cell wall as a result of adaptation to the surrounding environment The cell wall is mainly composed of chitin, β-glucan, and mannoproteins [52]. Among these components, the most abundant proteins are those with GPI anchor-binding sequences (88%), followed by PIR–cell wall proteins [32]. On the other hand, cell membranes play a fundamental role in maintaining cellular integrity. They act as a highly selective permeability barrier between the intracellular and extracellular environments and serve as a binding point for the cytoskeleton. Additionally, cell membranes are mainly composed of lipids and proteins [51]. The main lipids in fungal membranes are glycerophospholipids, lysolipids, sphingolipids, and sterols, and the exact composition depends on the fungal species and strains [43]. The most abundant sterol in fungal membranes is ergosterol; for *C. albicans*, this comprises more than 50% of all sterols [53,54]. The cell membrane plays a crucial role not only in virulence and resistance to antifungals but also potentially, alongside the cell wall, in the response to oxidative stress [42]. Given the significance of the cell wall and membrane components, they represent promising targets for the discovery and development of new drugs [52].

Another target of the R-1-R peptide and the *B. pilosa* extract in the *C. albicans* membrane consists of transport proteins. This study confirmed that the R-1-R peptide reduces the activity of ABC-type efflux pumps, especially Cdr1/Cdr2 (Figure 9). However, this effect is not directly related to the candidacidal activity, as no changes in susceptibility are evident in the mutant strains *cdr1∆/CDR1* and *cdr2∆/CDR2* (Table 1). Regarding the *B. pilosa* extract, it is suggested that it may activate efflux pumps (Table 1, Figure 9) possibly due to basal glucose content in this crude extract. Nevertheless, further experiments are required to confirm this effect.

Damage to the cell wall can generate stress in the yeast, triggering responses to regenerate the damage caused by an external agent [32,55]. This is evidenced by the up-regulation of cell wall biogenesis proteins in both study strains (Figure 12). The response to cell wall stress was confirmed using mutant strains and the proteomic study, where it was evident that with the R-1-R peptide, the mutant strains of MAPKs, Mkc1, (specific for wall stress cellular [32]), and Hog1 (core protein responsive to different types of stress [48]) were highly more susceptible to R-1-R. On the other hand, in the 256 strain, when it was treated with the R-1-R peptide, the proteins Bck1 (important in the MAPKs pathway, upstream of Mck1) and Ssk2 (upstream of Hog1) [32] were down- and up-regulated, respectively (Figure 13A).

Similar to the effects observed with R-1-R and the *B. pilosa* extract on the cell wall and membrane, these substances can also disrupt the mitochondrial membrane (Figure 13B), which is an important source of ROS and impacts the transport of electrons in the respiratory chain, thus generating mitochondrial dysfunction. These effects have been noted with commercial antifungal agents [43].

R-1-R and, to a greater extent, the extract of *B. pilosa* as well as their combination induce significant mitochondrial dysfunction, as demonstrated by mitochondrial membrane depolarization (Figure 11). This effect was also observed with the Ali1 and Cox4 mutant strains, which displayed varying sensitivities to the peptide or the extract. Figure 13C shows that R-1-R regulates proteins related mainly to cytochrome C in both study strains, affecting enzyme Q and complexes III and IV of the ETC. In contrast, the combination impacts complexes I to III and even the final stage of ATP synthesis. These observations suggest that damage to the ETC may lead to the excessive release of ROS, thereby causing oxidative stress in the yeast.

As widely recognized, excessive ROS can oxidize important cellular molecules such as DNA, lipids, and proteins. If this damage is severe, it can lead to cell death through necrosis, as observed in some cells treated with the extract or combination in SC5314 (Figure 7e,f) or with all treatments in 256 (Figure 8). However, prior to necrosis and as part of the adaptation to oxidative stress generated by the immune response of the host, *C. albicans* exhibits responses against oxidative stress in which Hog1p and Cap1p play important roles [48,55].

The proteomic approach revealed that parallel to the Hog1 pathway, another oxidative stress pathway was being activated, the pathway dependent on Cap1p (Figure 13D), where it was observed that two proteins downstream of Cap1p, Ypb1p and Gpx2p [56], were up-regulated. Additionally, for both study strains, the up-regulation of antioxidant proteins such as Sod1 and Sod4 and a stress response protein located in peroxisomes was evident.

The 2,7-Dichorofluorescein Diacetate fluorescent probe assay allowed for the confirmation of oxidative stress following the increase in the accumulation of intracellular ROS (Figure 10), which was caused by the peptide R-1-R. This presence of ROS was associated with the candidacidal effect of the peptide. as pretreatment with an antioxidant increased the tolerance of SC5314 to R-1-R. Regarding the *B. pilosa* extract, studies are needed to confirm the masking of the increase in ROS, which was caused by the antioxidant metabolites present in the extract, an effect that is replicated for the combination. 

Furthermore, it was evident that as a survival mechanism, the two study strains triggered programmed cell death signals, which evidenced the up-regulation of proteins present in autophagy and mitophagy (Figure 13E). Additionally, it was observed that at the nucleic acid level, for both strains, DNA preparation proteins (due to oxidative stress) and ribosomal biogenesis are being regulated (Figure 13F). Finally, it can be suggested that in cases where damage due to cell wall stress or oxidative stress was severe, a necrotic process was triggered (Figure 13G). 

## 4. Materials and Methods

### 4.1. Peptide

The peptide R-1-R was synthesized via solid-phase peptide synthesis using the Fmoc/tBu strategy, purified using RP-SPE chromatography, and characterized by means of RP-HPLC and MS according to the protocols reported in [15,17,57,58].

### 4.2. B. pilosa Extract

*B. pilosa* leaves obtained from Mocoa (Putumayo, Colombia) were used to carry out an extraction via maceration with 96% EtOH following the methodology reported in [13]. A stock solution of *B. pilosa* ethanolic extract was dissolved in dimethyl sulfoxide (DMSO) at a final concentration of 80 mg/mL and stored at 20 °C. In all experiments, a DMSO control was included.

### 4.3. Candida Strains

The reference strain *C. albicans* SC5314 and *C. albicans* 256 PUJ-HUSI, the FLC-resistant isolate, were previously characterized and identified using matrix-assisted laser desorption/ionization–time-of-flight mass spectrometry (MALDI TOF-MS) in our group [13,17]. Nine heterozygous or homozygous mutant strains from a *C. albicans* library [31] were used; the mutant strains are detailed in Table 1. The strains were grown on Sabouraud Dextrose Agar (SDA) plates, incubated overnight at 37 °C, and stored at 4 °C until use.

### 4.4. Antifungal Activity Assays with Mutant Strains 

The MICs were determined following the guidelines of the Clinical Laboratory Standards Institute (CLSI BMD- M27-A3) [59,60]. Serial dilutions of the treatments were made—for peptides from 6.25 to 200 µg/mL, for extracts from 31.2 to 2000 µg/mL, and for FLC from 0.025 to 32 µg/mL—with Roswell Park Memorial Institute (RPMI) 1640 medium in a 96-well microdilution plate (100 µL); then, 100 µL of the adjusted inoculum (0.5–2.5 × 10^3^ cells/mL) was added, incubated at 37 °C, and visual and spectrophotometric readings (595 nm) in the iMarKTM microplate reader (Bio-Rad, Hercules, CA, USA) were performed at 48 h. For the controls, growth control was conducted in RPMI or DMSO, as sterility controls correspond to RPMI medium and saline solution. The MIC was defined as the minimum concentration of treatment that inhibits 50% of growth compared to the control. To determine the MFC from the concentrations where no growth was observed, including the controls, a subculture was made on SDA (Difco) and incubated for 24 h at 37 °C. The MFC was defined as the minimum concentration of treatment that eliminates 100% of growth compared to the control. All tests were performed in triplicate.

### 4.5. Sample Preparation for Proteomic Analysis 

Yeast cells (1 × 10^7^ cell/mL) were pretreated in RPMI 1640 medium (with MOPS, pH 7.2) for 6 h at 37 °C as follows: *C. albicans* SC5314 was treated with the peptide R-1-R (CMI 100 µg/mL) and a combination of R-1-R (25 µg/mL) and an ethanolic extract of *B. pilosa* leaves (250 µg/mL), where a synergistic effect was previously found. On the other hand, *C. albicans* 256 was treated only with R-1-R (CMI 100 µg/mL). The concentration values of the evaluated substances used were previously reported in [13]. The treated cells were washed twice with phosphate-buffered saline (PBS). After, the cells were resuspended with lysis buffer [50 mM Tris HCl pH 7.5, 1 mM Ethylenediaminetetraacetic acid (EDTA), 150 mM NaCl, 1 mM dithiothreitol (DTT), 1 mM phenylmethylsulfonyl fluoride (PMSF), and protein inhibitor]; later, glass beads (0.5–0.78 mm) were added, and the sample was taken to a FastPrep equipment (MP Biomedicals, Santa Ana, CA, USA) (8 cycles of 20 s at 5.5 speed and cooling on ice, with 5 min between cycles). Once the extract of the cytoplasmic proteins was obtained, this was separated from the glass beads via centrifugation (13,000 rpm for 20 min), and the supernatant was collected and clarified via centrifugation [61]. The protein concentrations were determined using a Bradford Assay, using bovine serum albumin as the standard. Then, 1D gel electrophoresis was performed to determine the homogeneity of the samples (10 µg of protein), and the samples were adjusted to 1ug. Four biological replicates were made. Digestion and desalting were carried out on the samples with the PreOmics^®^’ iST Kit according to the protocol of the supplier [62]. 

### 4.6. Proteomic Analysis 

The peptides were analyzed using RP-LC-ESI-MS/MS using an EASYnLC1000 System (Thermo Scientific, Mississauga, ON, Canada) coupled to a Q-Exactive-HF mass spectrometer via a Nano-Easy spray source (Thermo Scientific, Mississauga, ON, Canada). The peptides were concentrated using reversed-phase (RP) chromatography using an Acclaim PepMap 100 precolumn (Thermo Scientific, 20mm × 75 µm ID, C18 3 µm particle diameter, and 100 Å pore size) with an integrated spray tip, operating at a flow of 250 nL/min. Subsequently, the peptides were eluted using a gradient from 5% to 35% of buffer B for 220 min, from 35 to 45% for 10 min, from 45 to 95% for 10 min, and 10 min at 95%. As buffer A, 0.1% AF with 2% ACN in water was used, and as buffer B, 0.1% formic acid in ACN was used. The entry of the peptides was carried out using electrospray ionization in positive mode, and they were analyzed in a Q Exactive HF mass spectrometer in DDA (data-dependent acquisition) mode. From each mass spectrum (MS) scan (between 350 and 1700 Da), the 10 most intense precursors (charge between 2+ and 5+) were selected for HCD (high collision-energy dissociation) fragmentation, and the corresponding MS/MS spectra were acquired.

The data files generated in the shotgun analysis were processed using Proteome Discoverer software (Thermo Scientific). The identification of peptide-spectrum matches (PSMs) of each MS/MS spectrum was performed by comparing them with theoretical mass lists derived from the *Candida* Genome Database (CGD) and the contaminant database. The peptides thus identified were assigned to their corresponding proteins. In cases where a peptide matched multiple proteins, the software employed the principle of parsimony to generate a “Master” protein, which was reported in the results. The percolator algorithm was used to estimate the False Discovery Rate (FDR), and the proteins identified with high confidence were filtered using a q value threshold of <0.01, following an adjusted protocol from [61].

### 4.7. Label-Free Quantification

Data normalization was performed using Protein Discoverer software. A correction factor was applied with the sum of the intensities of all the peptides for each of the samples, and peptide ratios were calculated as the median of the sum of the abundances for each replicate of each condition. The protein ratio was then determined as the geometric median of the peptide group ratios. For statistical analysis, the Analysis of Variance (ANOVA) was used to estimate the probability of differences between conditions. Significant comparisons were identified using T-student with a *p*-value threshold of *p* < 0.05. This analysis was refined using the Benjamin–Hochberg ad hoc test, achieving an adjusted *p*-value, also called q-value, with a threshold of q < 0.05 for a better control of the FDR. The filters applied to obtain the list of differential proteins (up- and down-regulated proteins) were (i) abundance ratio variability, where the variability of the ratios calculated as a percentage must present a value below 30% and above 0%; this last case ensured the elimination of the proteins that had been quantified from a single measurement, (ii) *p*-value <0.05, and (iii) q-value < 0.05 [63].

### 4.8. Bioinformatics Analysis

Bioinformatics analysis was conducted on the differential proteins in the following comparisons: R-1-R peptide vs. basal SC5314, combination vs. basal SC5314, and R-1-R peptide vs. basal 256. Initially, a functional analysis was performed to determine whether specific biological functions and known processes were over-represented in the dataset. This analysis was conducted using the “gprofiler2” package of RStudio [64], which facilitates functional analysis from a list of protein identifiers using the Ensembl database. The parameters used in the functional analysis were as follows: reference organism, with “*Candida albicans*” as the reference organism for the list of protein identifiers, *p* value < 0.05, and multiple-testing correction method, which was the FDR. The databases used for the enrichment of biological terms were Gene Ontology (covering the molecular function (MF), cellular component (CC), and biological process (BP)), Kyoto Encyclopedia of Genes and Genomes (KEGG), Reactome (REAC), and WikiPathways (WP). The results are represented using bar plots. To predict the protein–protein interaction networks, the STRING database https://string-db.org/ (accessed on 9 February 2024) was used. Finally, the KEGG database https://www.genome.jp/kegg/ (accessed on 12 February 2024) was used for protein pathway annotation.

### 4.9. Scanning Transmission Electron Microscopy (STEM)

Yeast strains (3 × 10^8^ cells/mL) were pretreated with subMIC concentrations of peptide, extract, or their combinations for 2 h. The solutions were then washed with PBS, followed by the addition of 100 µL of glutaraldehyde 2.5%, and left at room temperature for 18 h. After this incubation period, the suspensions were washed again with PBS. The pellets were fixed, desiccated, and embedded as previously described [65]. Then, the cells were observed using a Tescan Lyra 3 scanning electron microscope (Tescan, Kohoutovice, Czech Republic) [66] at the University of the Andes.

### 4.10. Efflux of Rhodamine 6G 

Yeasts cells were grown in 10 mL of yeast peptone dextrose (YPD) broth at 35 °C for 20 h (shaking at 150 rpm). After, 2.5 mL of the suspension was added to 22.5 mL of PBS (1–5 × 10^7^ cells/mL) and incubated at 37 °C for 2 h. Subsequently, the cells were centrifuged at 3000× *g* for 5 min. The pellets were washed three times with sterile distilled water, and they were suspended in a solution of Rhodamine 6G (R6G) 83697 (Sigma-Aldrich St. Louis, MO, USA) (20 µM), and the peptide or *B. pilosa* extract was added. The volume was adjusted to 25 mL with PBS. After the incubation, the cells were washed twice (3000× *g*; 5 min; 4 °C) with sterile distilled water, and the pellet was suspended in 25 mL of PBS (4 °C). Then, 5 mL of the suspension was added to a new centrifuge tube with 500 µL of glucose (final concentration of 2 mM). As a negative control, instead of glucose, 500 µL of PBS was added. After, in intervals of 0, 30, 60, 90, 120, and 150 min, 400 µL of the cell suspension was transferred to a microcentrifuge tube, and cells were collected via centrifugation at 10,000× *g* for 1 min (4 °C). Then, 100 µL of supernatant was added to a 96-well plate (dark). Finally, the fluorescence of the released R6G was measured using a Varioskan LUX spectrofluorometer (Thermo Scientific™, Waltham, MA, USA) at an excitation wavelength of 530 nm and emission wavelength 560 nm. Using the R6G calibration curve, the fluorescence intensity was converted to concentration. Three technical replicates were made [37,67].

### 4.11. Measurement of Intracellular ROS

Intracellular ROS levels were measured using 2,7-Dichorofluorescein Diacetate (DCFH-DA) (Merck-Millipore). Here, 1 × 10^7^ cells/mL, were treated with or without different concentrations of peptide, extract, or combinations for 2 h. Subsequently, the cells were washed with PBS, resuspended in PBS containing 20 µg/mL of DCFH-DA, and incubated at 30 °C for 30 min. After this time, the fluorescence intensity was measured with a Varioskan LUX spectrofluorometer (530 nm excitation wavelength and 560 nm emission wavelength). To determine the connection between ROS generation and the effectiveness of the peptide, extract, or combination against *C. albicans*, an ROS scavenger was used: N-Acetyl-L-cysteine (NAC) (Sigma-Aldrich). For this, 1 × 10^7^ pretreated cells/mL were incubated at 37 °C with NAC (60 mM) for 1 h. Subsequently, the cells were washed with PBS, and the fluorescence intensity was measured as described above. AmB at concentrations of 4 μg/mL and 64 μg/mL was utilized as a positive control for *C. albicans* SC5314 and *C. albicans* 256, respectively [25,46,68].

### 4.12. Measurement of the Mitochondrial Membrane Potential 

The mitochondrial membrane potential was measured using Rhodamine (Rho) 123 staining (Sigma-Aldrich) according to [46]. *C. albicans* cells (1 × 10^7^ cells/mL) were exposed to different concentrations of peptide, extract, or combinations for 2 h at 35 °C. After treatment, the cells were washed twice with PBS and resuspended in a solution containing 25 µM Rho123. This suspension was incubated at 35 °C for 20 min. After washing three times, the fluorescence intensity was measured using a Varioskan spectrofluorometer LUX (Thermo Scientific™, Waltham, MA, USA) at an excitation wavelength of 480 nm and an emission wavelength of 530 nm. Cells without treatment and those incubated with sodium azide (NaN_3_) (Sigma-Aldrich) (5 mM) for 20 min were used as negative and positive controls, respectively.

### 4.13. Statistical Analysis

The experiments were carried out with a minimum of three biological replicates and at least two technical replicates. Data are presented as means ± standard deviation (SD) and were analyzed using an ANOVA multiple comparisons test. A *p*-value of <0.05 was considered significant. The statistical models were achieved using GraphPad software (version 7) (GraphPad Software Inc., La Jolla, CA, USA). 

## 5. Conclusions

Our findings indicate that using mutant strains along with proteomics and biological assays can be a viable strategy for studying mechanisms of action. However, complementary research is necessary to determine whether the observed effects of the molecules align with the mechanism of action or if they derive from a secondary response to these mechanisms. In this study, we confirmed that for R-1-R, similar to other LfcinB derivatives, the candidacidal mechanism is closely related to damage to the cell wall and membrane as well as oxidative stress induced by mitochondrial dysfunction. Regarding the extract of *B. pilosa*, this is the first study on the mechanisms of action in vitro for extracts of this plant, which allowed us to demonstrate the impact of biological processes similar to the R-1-R peptide, although at different levels and possibly in a milder way. When combined with R-1-R, the extract significantly enhances the damage to yeast at low treatment concentrations.

Finally, it was possible to show that the peptide, extract, and the combination exert antifungal activity against *C. albicans*, regardless of the resistance phenotype. However, we found that resistance to FLC due to mutations in the ergosterol biosynthesis pathway increases the susceptibility to oxidative stress damage. This study provides a foundation for understanding the complex antifungal effects of natural product derivatives and represents a step forward in elucidating their mechanisms of action.

## Figures and Tables

**Figure 1 ijms-25-08938-f001:**
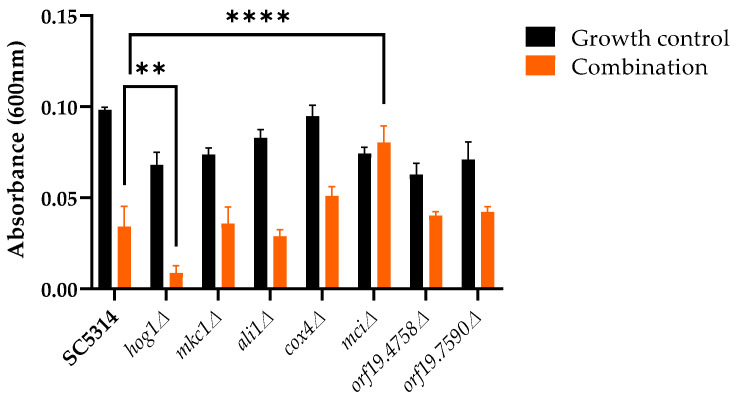
Susceptibility of homozygous mutant strains after incubation with the combination of R-1-R peptide and *B. pilosa* extract. Data are means ± SD from three experiments. ** *p* < 0.01, and **** *p* ≤ 0.0001.

**Figure 2 ijms-25-08938-f002:**
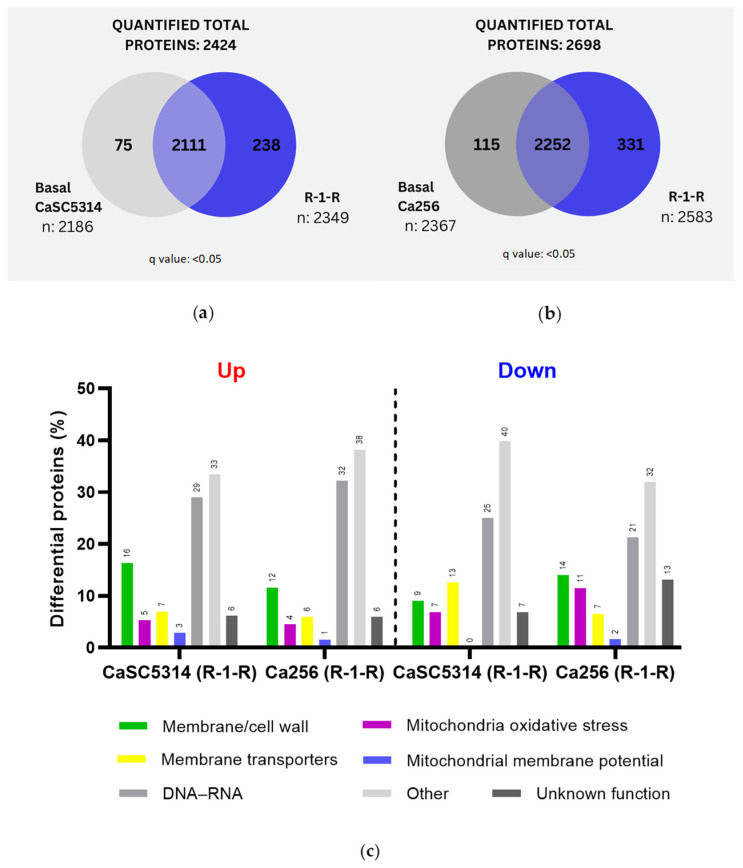
Venn diagram of the total and exclusive proteins obtained after treatment with R-1-R in (**a**) SC5314 and (**b**) 256. (**c**) Up- and down-regulated proteins in SC5314 and 256 treated with R-1-R, grouped according to their biological function.

**Figure 3 ijms-25-08938-f003:**
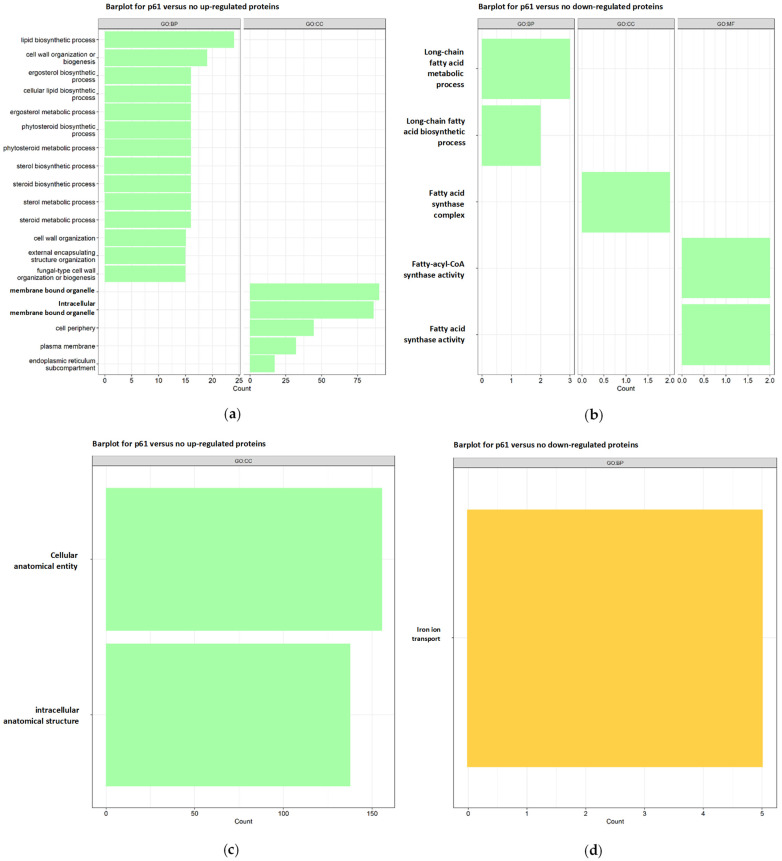
Bar plots of functional analyses for (**a**) up- and (**b**) down-regulated proteins for strain SC5314 and (**c**) up- and (**d**) down-regulated proteins in strain 256 treated with R-1-R. The green bars represent terms associated with the membrane or cell wall, yellow bars represent terms associated with membrane transporters, dark gray bars are terms associated with the nucleus or DNA/RNA, and light gray bars are the terms associated with other cellular processes.

**Figure 4 ijms-25-08938-f004:**
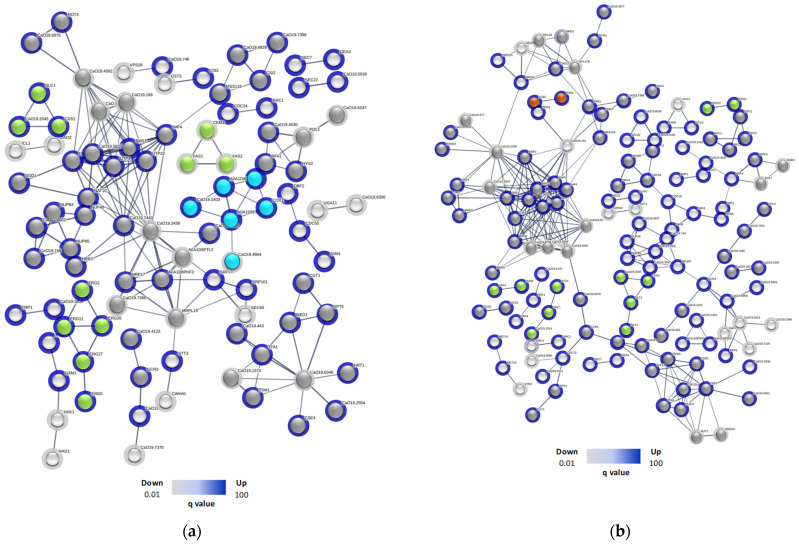
The STRING protein–protein interaction network of up- and down-regulated proteins in (**a**) SC5314 and (**b**) 256 after treatment with R-1-R. The nodes of the network are proteins identified by gene name, while the lines are edges that represent functional associations based on different types of evidence. The haloes of the nodes, in blue, belong to up-regulated proteins and the gray halos to down-regulated ones. The colors in the nodes represent biological processes, such as those related to the membrane (green), nucleic acids (dark gray), mitochondrial membrane potential (light blue), autophagy and endocytosis (orange), and others (light gray).

**Figure 5 ijms-25-08938-f005:**
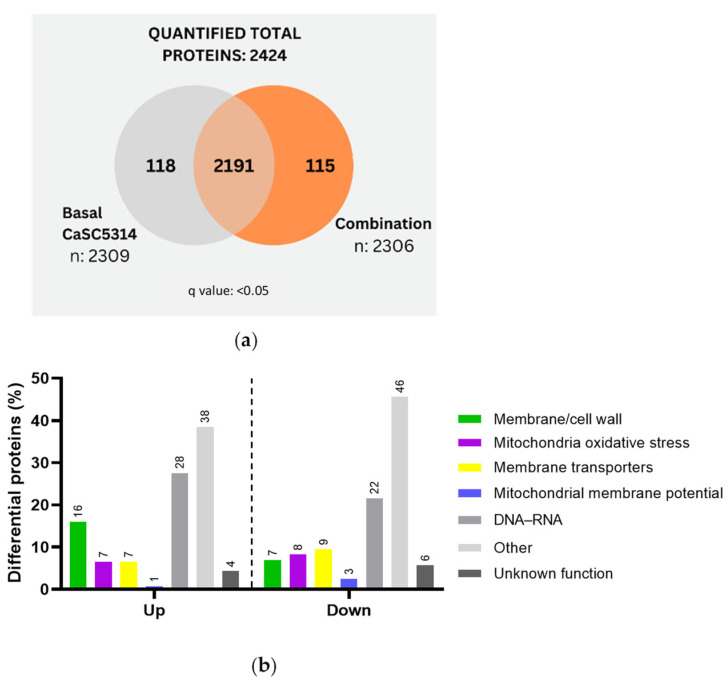
(**a**) Venn diagram of the total and exclusive proteins obtained in SC5314 after treatment with the combination between R-1-R and *B. pilosa* extract. (**b**) Up- and down-regulated proteins in SC5314 treated with the combination, grouped according to their biological function.

**Figure 6 ijms-25-08938-f006:**
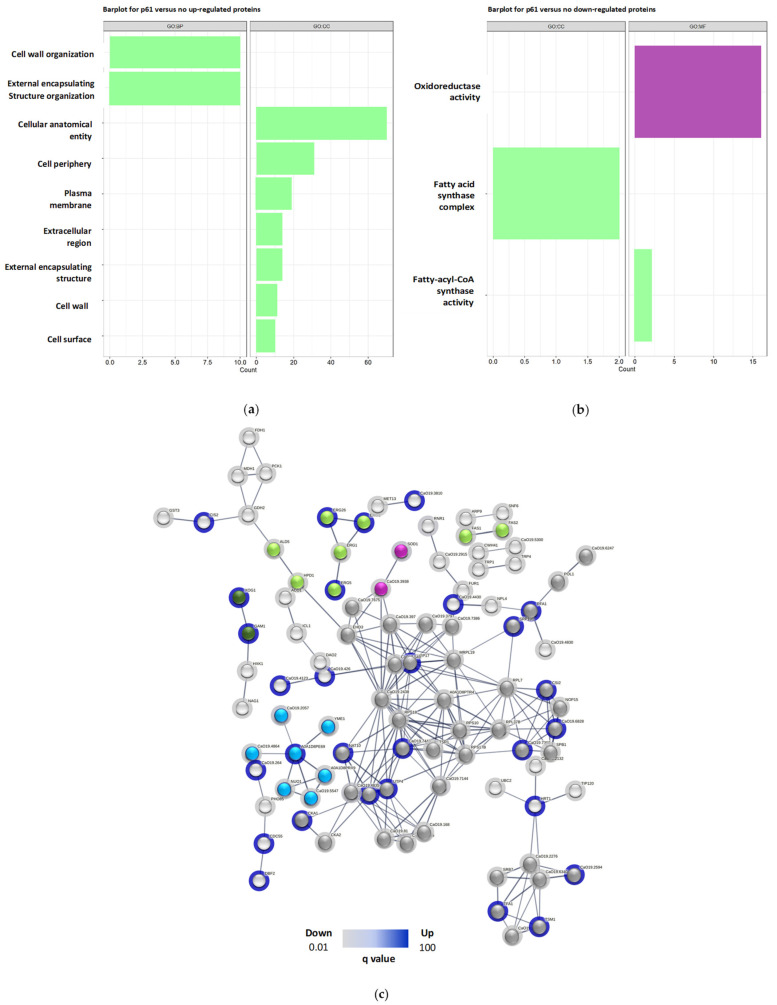
Bar plots of functional analyses for (**a**) up- and (**b**) down-regulated proteins for strain SC5314 treated with the combination. (**c**) The STRING protein–protein interaction network of SC5314 treated with the combination. The nodes of the network are proteins identified by gene name; the lines are edges that represent functional associations based on different types of evidence. The haloes of the nodes, in blue, belong to up-regulated proteins and the gray halos to down-regulated ones. The colors in the nodes represent biological processes, such as those related to the membrane (green), oxidative stress (lilac), mitochondrial respiration (blue), nuclide acids (dark gray), and others (light gray).

**Figure 7 ijms-25-08938-f007:**
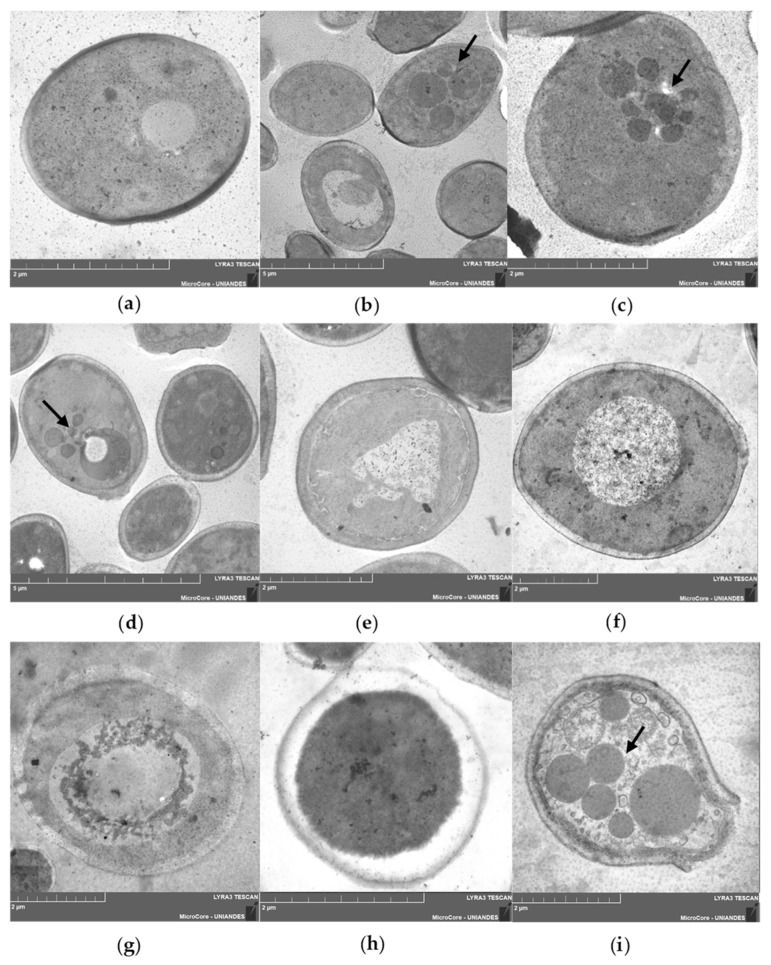
STEM images of *C. albicans* SC5314 without treatment (**a**) and after 2 h of incubation with R-1-R (**b**,**c**), *B. pilosa* extract (**d**,**e**), or a combination of peptide and extract (**f**–**i**), where there was a synergistic effect; R-1-R (25 µg/mL) and extract *B. pilosa* (250 µg/mL). The arrows indicate the cytoplasmic microbodies.

**Figure 8 ijms-25-08938-f008:**
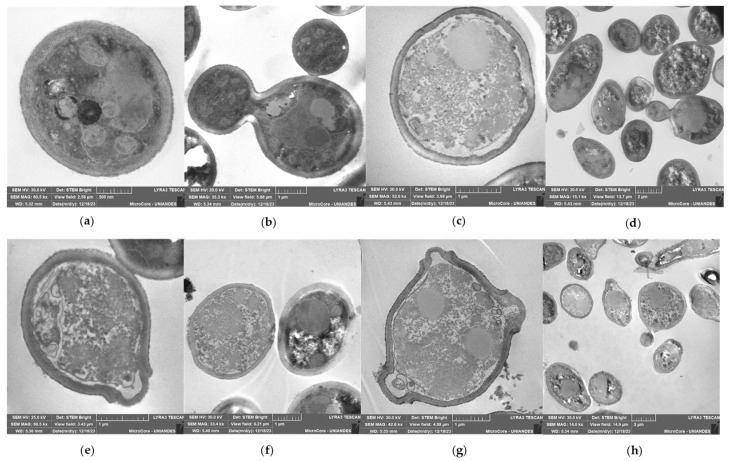
STEM images of *C. albicans* 256 without treatment (**a**,**b**) and after 2 h of incubation with R-1-R (**c**,**d**), *B. pilosa* extract (**e**,**f**), or a combination of peptide and extract (**g**,**h**), where there was a synergistic effect; R-1-R (50 µg/mL) and extract *B. pilosa* (62.25 µg/mL).

**Figure 9 ijms-25-08938-f009:**
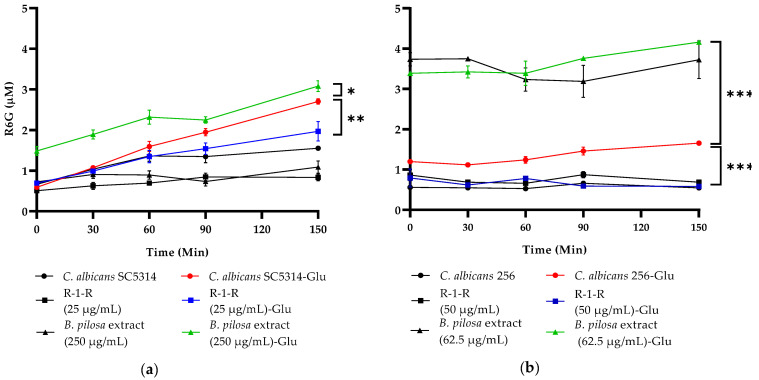
Rhodamine 6G (R6G) efflux over time in (**a**) *C. albicans* SC5314 and (**b**) *C. albicans* 256. Colored lines indicate the concentration of R6G released after the addition of 20 mM glucose (Glu); black lines correspond to the controls without Glu. Data are means ± SD from three experiments. * *p* < 0.05, ** *p* < 0.01, and *** *p* < 0.001.

**Figure 10 ijms-25-08938-f010:**
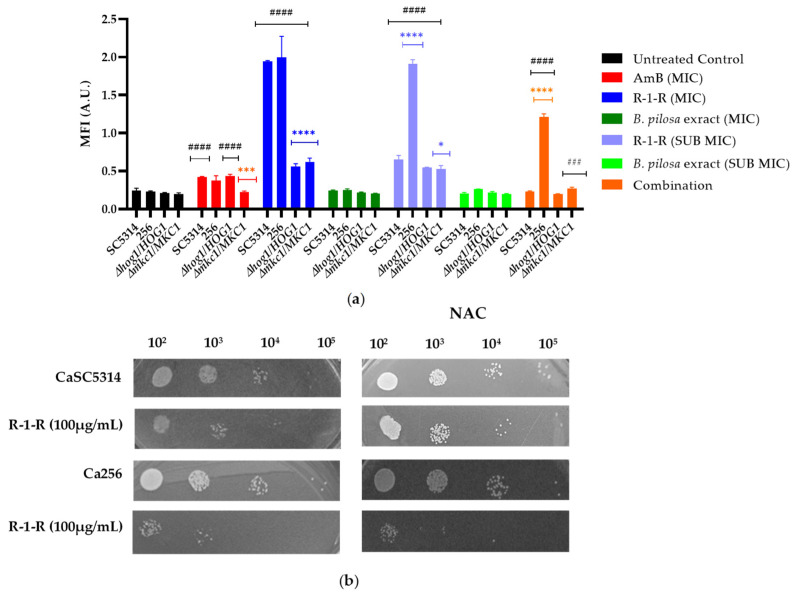
(**a**) ROS accumulation on *C. albicans*. ROS production was assessed via staining with H2DCFDA; cells were pretreated with varying concentrations of R-1-R, (SC5314 and mutant strains: MIC: 100 µg/mL, SUB MIC: 25 µg/mL; 256: MIC: 100 µg/mL, SUB MIC: 50 µg/mL) and extract (SC5314 and mutant strains: MIC: 500 µg/mL, SUB MIC: 250 µg/mL; 256: MIC: 500 µg/mL, SUB MIC: 62.5 µg/mL), and Amphotericin B (SC5314 and mutant strains: 4 µg/mL; 256: 64 µg/mL) was used as the positive control. MFI, mean fluorescence intensity; A.U., arbitrary units. The results were obtained as means ± SD from three experiments. * *p* < 0.05, *** *p* < 0.001, and **** *p* ≤ 0.0001 compared with SC5314 in each group treated; ^###^
*p* < 0.001, and ^####^
*p* ≤ 0.0001 compared with SC5314 without treatment. (**b**) The effect of ROS scavenger, NAC, on the candidacidal activity of R-1-R (100 µg/mL) was also examined.

**Figure 11 ijms-25-08938-f011:**
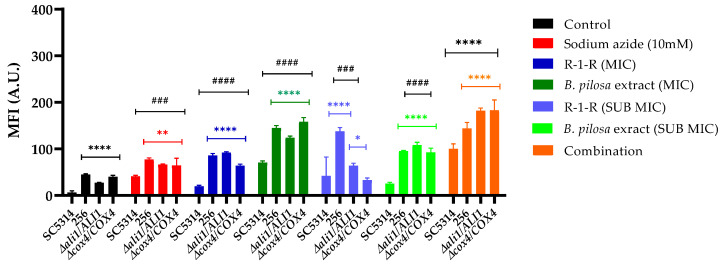
Mitochondrial membrane potential was measurement using rhodamine 123 staining. Cells were treated with R-1-R, (SC5314 and mutant strains: MIC: 100 µg/mL, SUB MIC: 25 µg/mL; 256: MIC: 100 µg/mL, SUB MIC: 50 µg/mL) and extract (SC5314 and mutant strains: MIC: 500 µg/mL, SUB MIC: 250 µg/mL; 256: MIC: 500 µg/mL, SUB MIC: 62.5 µg/mL), and sodium azide (NaN3; 5 mM) was used as the positive control. MFI, mean fluorescence intensity; A.U., arbitrary units. The results were obtained as means ± SD from three experiments. * *p* < 0.05, ** *p* < 0.01, and **** *p* ≤ 0.0001 compared with SC5314 in each group treated; ^###^
*p* < 0.001, and ^####^
*p* ≤ 0.0001 compared with SC5314 without treatment.

**Figure 12 ijms-25-08938-f012:**
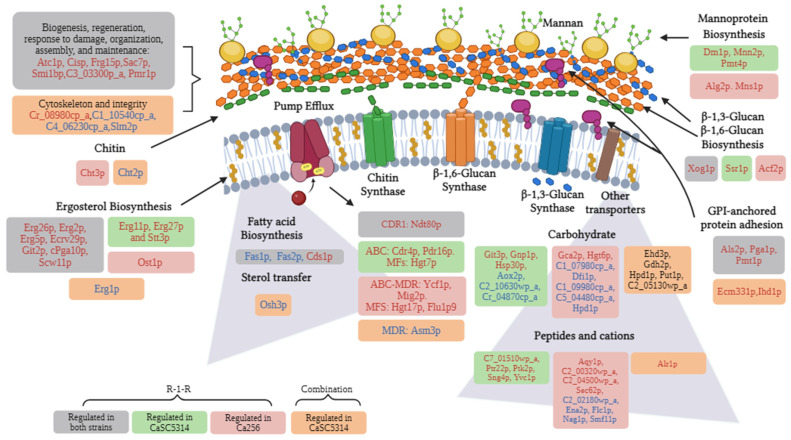
Scheme of the up-regulated (red letters) and down-regulated (blue letters) proteins by the action of the R-1-R peptide and its combination (orange bubbles) with an extract of *B. pilosa* against the cell wall and membrane of both strains of *C. albicans* SC5314 and *C. albicans* 256 (gray bubbles) or exclusively in each strain [*C. albicans* SC5314 (green bubbles) and *C. albicans* 256 (red bubbles)].

**Figure 13 ijms-25-08938-f013:**
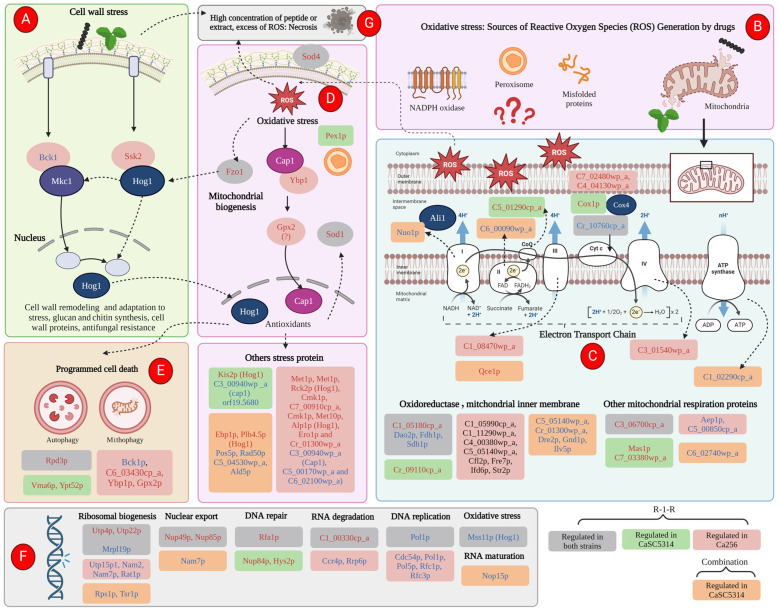
Scheme of the up-regulated (red letters) and down-regulated (blue letters) proteins by the action of the R-1-R peptide and its combination (orange bubbles) with an extract of *B. pilosa against* both strains (gray bubbles) of *C. albicans* SC5314 (green bubbles) and *C. albicans* 256 (red bubbles) or exclusively in each strain.

**Table 2 ijms-25-08938-t002:** Functional categories of *C. albicans* SC5314 and *C. albicans* 256 proteins regulated by R-1-R.

*C. albicans* SC5314				*C. albicans* 256		
Systemic Name	Function	Regulation Type		Systemic Name	Function	Regulation Type
**Cell wall**				**Cell wall**		
C1_11270wp_a		Up		Acf2p	Endo-1,3-beta-glucanase	Up
Dpm1p	Mannoprotein biosynthesis	Up		Ada2p	Integrity	Up
Gda1p	Cell wall and cell surface charge	Up		Alg2p	Mannoprotein biosynthesis	Up
Mnn2p	Mannoprotein biosynthesis	Up		C2_08580wp_b		Up
Pmt4p	Mannoprotein biosynthesis	Up		C2_10740cp_a	Integrity	Up
Rbe1p	Regulator of cell wall dynamics	Up		Cht3p	Chitin	Up
Sim1p	Maintenance	Up		CHrr25p	Stress	Up
Spf1p	Formation	Up		Mns1p	Mannoprotein biosynthesis	Up
Ssr1p	Beta-glucan	Up		Cwt1p		Up
Atc1p	Formation	Up		Rom2p	Biogenesis	Up
Als2p	Regeneration	Up		Sac1p	Integrity Integrity	Up
Cis2p	Biogenesis	Up		Bck1p		Down
Fgr15p	Damage response	Up		Evp1p	Organization	Down
Sac7p	Organization	Up		Pde2p		Down
Smi1bp	Assembly	Up		Msb2p	Cell wall stress	Down
C3_03300cp_a		Up		C1_10540cp_a	Cytoskeleton	Up
Pmr1p	Maintenance	Up		Cr_08980cp_a		Up
Xog1p	Exo-1,3-beta-glucanase-chitin	Up		C5_04890cp_a		Down
Ecm15p	Organization	Down		Nce102p	Cytoskeleton, actin	Down
Rhb1p	Integrity	Down				
**Cell membrane**				**Cell membrane**		
Erg11p	Ergosterol biosynthesis	Up		Dnf1p	Sphingolipid translocation	Up
Erg27p		Up		Sct1p	Phospholipid biosynthesis	Up
Stt3p		Up		Stt4p	Kinase	Up
Erg26p	Ergosterol biosynthesis	Up		C1_03690wp_a	Ergosterol biosynthesis	Up
Erg2p		Up		Cht3p	Biosynthesis+C39:C44 ergosterol	Up
Erg5p		Up		Sld1p	Ergosterol biosynthesis	Up
Ecrv29p		Up		Ost1p		Down
Git2p		Up		C1_02270cp_a	Fatty acid catabolism	Down
cPga10p		Up		Ece1p	Integrity	Down
Scw11p		Up				
C6_03240wp_a		Down				
C3_00570cp_a	Phosphoprotein	Up				
Cds1p	Phospholipid biosynthesis	Up				
Ino1p	Inositol-3-phosphate	Up				
Osh2p	Sterol transfer	Up				
C2_05290cp_a	Component fatty acids	Up				
C2_10010cp_a		Up				
C5_05440cp_a	Phospholipid binding	Up				
Cdc24p		Up				
C5_01420wp_a		Up				
Fmp45p		Up				
Rac1p	G-protein of RAC subfamily	Up				
Slc1p	Glycerolipid biosynthesis	Up				
Fas1p	Biosynthesis fatty acids	Down				
Fas2p		Down				
Icl1p	Catabolism fatty acids	Down				
**Mitochondria, oxidative stress**				**Mitochondria, oxidative stress**		
C5_01290cp_a	Coenzyme Q biosynthesis Oxidoreductase	Up		Met1p	Oxidative stress, Hog1	Up
Cr_09110cp_a		Up		Rck2p		Up
Kis2p	Oxidative stress, Hog1	Up		Ssk2p		Up
Mas1p	Mitochondrial respiration Peroxisome	Up		C7_00910cp_a	Oxidative stress, Cap1	Up
Pex1p		Up		Cr_01300wp_a	Oxidative stress	Up
C1_05180cp_a	Oxidative stress, Cap1	Up		Cmk1p		Up
Fzo1p	Oxidative stress, Hog1	Up		Met10p	Reductase, Hog1 induced	Up
C1_05180cp_a	Oxidoreductase antioxidant	Up		Alp1p		Up
Cr_09110cp_a	Oxidoreductase activity Oxidative stress response to ROS	Up		Ero1p	Oxidoreductases, homeostasis	Up
Sur2p		Up		C5_00170wp_a	Oxidative stress	Down
Sod1p	Oxidoreductase activity Oxidative stress	Up		C6_02100wp_a		Down
Sod4p		Up		C1_05990cp_a	Oxidoreductases	Down
C3_00940wp_a	Oxidoreductase activity	Down		C1_11290wp_a		Down
C5_00170wp_a		Down		C4_00380wp_a		Down
C5_02690wp_a	Oxidoreductase activity	Down		C5_05140wp_a		Down
Dao2p		Down		Cfl2p		Down
Fdh1p		Down		Fre7p		Down
Sdh1p		Down		Ifd6p		Down
				Str2p		Down
				C3_00940wp_a	Oxidative stress, Cap1	Down
**Mitochondrial membrane potential**				**Mitochondrial membrane potential**		
Cox1p		Up		C7_02480wp_a	Respiration, cytochrome C	Up
C3_06700cp_a	Respiratory chain	Up		C4_04130wp_a		Up
C7_03380wp_a		Up		C1_08470wp_a	Mitochondrial respiration, complex III	Up
C3_00620cp_a	Respiratory chain, Cytochrome C	Up		C3_01540wp_a	Mitochondrial respiration, complex IV	Up
C5_02740wp_a		Up		C5_02590cp_b	Respiration	Up
Cr_10760cp_a	Cytochrome C	Down		Mss116p		Up
				Aep1p		Down
				C5_00850cp_a		Down
**Transport**				**Transport**		
Cdr4p	ABC transporters	Up		Ycf1p	ABC-MDR transporters	Up
Pdr16p	Activates CDR1/CDR2	Up		Mig2p		Up
Ndt80p	Activates CDR1	Up		Hgt17p	MFS transporters	Up
Hgt7p	MFS transporters	Up		Flu1p		Up
Git3p	Carbohydrate transport	Up		Gca2p	Carbohydrate transport	Up
Gnp1p		Up		Hgt6p		Up
Hsp30p		Up		C1_07980cp_a		Down
Hgt19p	Carbohydrate transport	Up		Dfi1p		Down
Hgt19p		Up		C1_09980cp_a		Down
Aox2p	Carbohydrate transport	Down		C5_04480cp_a		Down
C2_10630wp_a		Down		Hpd1p		Down
Cr_04870cp_a		Down		Aqy1p	Water canal	Up
C1_09980cp_a	Carbohydrate transport	Down		C2_00320wp_a		Up
C5_04480cp_a		Down		C2_04500wp_a	Ion	Up
Sfc1p		Down		Sec62p	Proteins	Up
C7_01510wp_a		Up		C2_02180wp_a	Metals	Down
Ptr22p	Peptides	Up		Ena2p	Potassium ion	Down
Ptk2p		Up		Smf11p	Metals	Down
Sng4p		Up				
C7_03590cp_a	Antiport	Up				
Ftr1p	Fe ion	Up				
Ftr2p	Fe ion	Up				
Pho84p	Cations	Up				
Vrg4p		Up				
Flc1p	FAD	Down				
Yvc1p	Calcium	Down				
Nag1p		Down				
**Vacuole–mitochondria connection**				**Mitochondrial cell death**		
Vma6p	Mitochondria binding complex	Up		Bck1p	Mitophagy	Down
Ypt52p		Up		C6_03430cp_a	Autophagy mitochondria	Up
Rpd3p	Autophagy mitochondria	Up		Ybp1p		Up
				Gpx2p		Up
**DNA–RNA**				**DNA–RNA**		
Nup84p	Repair DNA damage due to oxidation	Up		Ccr4p	RNA degradation	Up
Hys2p		Up		Rrp6p		Up
Utp4p	Ribosome biogenesis	Up		Cdc54p	Replication	Up
Utp22p		Up		Pol1p		Up
Nup49p		Up		Pol5p		Up
Nup85p	Nuclear export	Up		Rfc1p		Up
Rfa1p	DNA damage repair	Up		Rfc3p		Up
C1_00330cp_a	RNA degradation	Up		Utp15p1	Ribosome biogenesis	Up
Pol1p	DNA Replication	Down		Nam2	tRNA biosynthesis	Down
Mrpl19p	Ribosome subunit	Down		Nam7p	Nucleocytoplasmic transport	Down
Mss11p	Transcription factor, MAPK signaling pathway (HOG1)	Down		Rat1p	Ribosome biogenesis	Down

Left table: unshaded cells correspond to differentially regulated proteins (up or down) exclusively in *C. albicans* SC5314. Light gray shaded cells correspond to differentially regulated proteins (up or down) in both *C. albicans* SC5314 and *C. albicans* 256. Right table: includes all differentially regulated proteins (up or down) exclusive to *C. albicans* 256.

**Table 3 ijms-25-08938-t003:** Functional categories of *C. albicans* SC5314 proteins regulated by the combination of R-1-R and *B. pilosa* extract.

Systemic Name	Function	Regulation Type
**Cell wall**		
Ecm331p	Cell surface GPI anchor	Up
Ihd1p		Up
Cr_08980cp_a	Cytoskeleton organization	Up
C1_10540cp_a	Cytoskeleton, actin	Down
C4_06230cp_a		Down
Slm2p		Down
Cht2p	Chitinase	Down
Ada2p	Wall integrity	Down
**Membrane cell**		
Erg1p	Ergosterol biosynthesis	Down
Osh3p	Sterol transfer	Down
**Mitochondria, oxidative stress**		
Ebp1p	Oxidative stress	Up
Plb4.5p	Oxidative stress, HOG1	Up
Pos5p	Oxidative stress	Down
Rad50p		Down
C5_05140wp_a	Oxidoreductases, mitochondrial matrix	Down
Cr_01300wp_a	Oxidoreductases, mitochondrial inner membrane	Down
Dre2p, Gnd1p	Oxidoreductases, redox homeostasis	Down
Ilv5p	Oxidoreductases	Down
C5_04530wp_a	Antioxidant activity	Down
Ald5p	Antioxidant activity, degradation of fatty acids	Down
**Mitochondrial membrane potential**		
Qce1p	Expression of complex III of the respiratory chain	Up
Nuo1p	Complex I of the respiratory chain	Down
C6_00090wp_a	Complex II of the respiratory chain	Down
C6_02740wp_a	Electron transport	Down
C1_02290cp_a	ATP synthesis	Down
**Transporters**		
Alr1p	Cation transport	Up
Ehd3p	Carbohydrate transport	Down
Gdh2p		Down
Hpd1p		Down
Put1p		Down
C2_05130wp_a	Carbohydrate transport, oxidoreductase	Down
Asm3p	MDR1 transporters	Down
**DNA–RNA**		
Rps1p	Ribosome	Down
Tsr1p		Down
Nop15p	rRNA maturation	Down
Nam7p	Nucleocytoplasmic transport	Down

## Data Availability

The mass spectrometry proteomics data have been deposited to the ProteomeXchange Consortium via the PRIDE [69,70] partner repository with the dataset identifier PXD053558 and 10.6019/PXD053558. Reviewer access details (non-publishable data). Log in to the PRIDE website using the following details: Project accession: PXD053558, Token: 7EWJZ1a4rW1T. Alternatively, reviewer can access the dataset by logging in to the PRIDE website using the following account details: Username: reviewer_pxd053558@ebi.ac.uk, Password: jAhLzUOT5Wxn.

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
