# Peer review of "Antifungal Synergy: Mechanistic Insights into the R-1-R Peptide and Bidens pilosa Extract as Potent Therapeutics against Candida spp. through Proteomics"

_ijms, 2024, doi:10.3390/ijms25168938_

Round 1

Reviewer 1 Report

Comments and Suggestions for Authors

This manuscript aims to understand the mechanism of action behind the antifungal activity of the peptide R-1-R and to provide insights on the antifungal synergy observed with the concomitant treatment of C. albicans with R-1-R and Biden pilosa extract. The authors provide several data obtained by proteomics, STEM, biochemical and phenotypic assays.

Specific comments:

Introduction:

To improve clarity, please consider changing the order by which some topics are described. For instance, you could explain first what the R-1-R peptide is, its origins and what is know so far on LfcinB, and only then refer to the reference [13]. Such information on LfcinB mechanism of action could be referred (also?) in the results chapter to justify why you are looking more carefully to certain processes.

Line 45: C. albicans IS an opportunistic pathogen and becomes pathogenic (rather than harmless commensal) in the conditions described. Remove “everything”.

Line 67-73: Beware that antifungal tolerance and resistance are not the same thing.

Line 113-114: “promote the aggregation of cytoplasmic” ??

Results:

I found the order in which the results are presented a little confusing. First, you analyse the susceptibility of several mutants to the peptide, extract and fluconazole, then you do proteomics without mentioning these proteins, and finally you go back to them to do more specific analysis on ROS and mitochondrial membrane potential. Maybe the proteomics and more general analysis should come first.

2.1:  As referred in the Introduction chapter (Line 70-73), resistance to fluconazole can be caused by several mechanisms. How do the authors know that Ca256 strain is resistant to fluconazole due to a mutation on ERG11 (line 143)? Please provide a reference.

 Is the info provided in table 1 a review of previously published data [13], which info is? Why there is no data on MIC and MFC (Table 1) for cells treated with both R-1-R and extract (antifungal combination)?

Please explain/discuss better why do you use heterozygous instead of homozygous mutants and the observed differences in MIC/MFC. Why weren´t CDR1/2 homozygous mutants also used?

Why were the MAPKs heterozygous mutants more resistant to FLC than the wild type? How do the null mutants behave?

Please explain better the hypothesis. This was the hypothesis for the results in paper [13], correct? But how does this explain the increased tolerance of mutants to the extract if FLC is not being simultaneously added?

Line 177-179: “…which suggests… decreases.” please remove the sentence, it is unnecessary.

Line 195: consider replacing “novel opportunity” for “alternative method” or something similar. The reference is from 2007.

Have you checked the role of Cek1 MAPK in this response?

Table S1: replace CMI by MIC

Figure S1: legend for c) and d)

Lines 227-236: improve. Have you done growth curves or measured the OD? It seems that the observed differences may be due to different growth rates. The drugs inhibit growth, thus less OD after 6h treatment, thus less protein. Are FLC resistant strains more fit? Please explain better how you obtained the samples for proteomic analysis (see Materials and Methods).

2.3: When comparing SC5314 and Ca256 there are more proteins identified in the resistant strain. Either common or exclusive…

Tables A from Appendix A.

Line 246-248: either the down-regulation and up-regulation of proteins could compensate for the damage... Please discuss this better. Why is the number of up-regulated and down-regulated proteins important? Did you analyse which specific proteins are having different behaviours?

Where did you get the information on column 6 “relationship with antifungal activity”?

Why don´t you show “others” as separate categories since these are the most represented?

Please revise the tables. For instance, Mitochondria – ox stress (purple) states as 13 entries but the purple cells also contain “others” and “unknown”.

2.4:  Which specific proteins change their expression when the extract is added (up-regulated with the peptide but down-regulated with the antifungal combination? I think it would be more interesting if comparisons were made directly between SC5314 + R-1-R and SC5314 + R-1-R + extract.

2.5: Why is the number of total SC5314 basal proteins in figure 4 different from figure 1?

2.6: Consider showing the list of all exclusively regulated proteins by the antifungal combination. How was the choice of that small subset of proteins made?

2.7: Please discuss better the results shown in Figure 8, particularly those seen for the extract treatment. Shouldn’t the values be normalized?

2.8:  Have you checked the phosphorylation state of the CaMAPKs in the presence of the peptide, extract and both?

Materials and Methods:

How was GO enrichment performed? Which software and which background (all Ca entries or only those proteins identified by MS?).

4.5: please explain better. Pre-inoculum in what, for how long at which temperature? Dilution in RPMI plus drug (or not) and incubated for 6h? Final OD measurements to correlate with protein concentration?

4.6: How did you obtain the peptides? Enzyme used, in-gel or in-solution digestion?

4.8: How did you determined overrepresentation?

Other comments:

Data availability: is the proteomic data available at PRIDE or are you planning to submit it to a public repository?

Comments on the Quality of English Language and Formatting

I found the manuscript confusing and difficult to follow primarily due to que quality of the English. It would benefit from a thorough language revision to improve its clarity and readability.

Line 3: “Bidens pilosa” in italic. Please review this in the entire manuscript including references.

Line 5-8: check format

Line 21-22: “studied” … ”study”… “study”..

Line 48: remove “causing”

Line 49: remove ";”

Line 131: “…a metabolite present in a…”

Line 209 (Table 1): Consider changing orf19 id for recent systematic name (Assembly 22); Please replace “CMI/CMF” for “MIC/MFC”; Oral clinical isolate, not “isolated”; …was obtained FROM [31]; Not determined.

Line 210-211: Format and remove the dot at proteins.

Line 759: according TO the protocol

Line 769: are the references correct?

Line 771: SDA meaning

Line 871: SDA?

Comments on the Quality of English Language

I found the manuscript confusing and difficult to follow primarily due to que quality of the English. It would benefit from a thorough language revision to improve its clarity and readability.

Author Response

Response to Reviewer 1 Comments

Comments and Suggestions for Authors

This manuscript aims to understand the mechanism of action behind the antifungal activity of the peptide R-1-R and to provide insights on the antifungal synergy observed with the concomitant treatment of C. albicans with R-1-R and Biden pilosa extract. The authors provide several data obtained by proteomics, STEM, biochemical and phenotypic assays.

Specific comments:

Introduction:

To improve clarity, please consider changing the order by which some topics are described. For instance, you could explain first what the R-1-R peptide is, its origins and what is know so far on LfcinB, and only then refer to the reference [13].

Answer: We appreciate the comment, with the aim of highlighting the combination between peptide and extract, as a therapeutic alternative, we have organized the introduction in this way, where we first refer to the achievement of improving the antifungal activity with the use of the combination and subsequently expand the detailed information on both the R-1-R peptide and the extract individually.

Such information on LfcinB mechanism of action could be referred (also?) in the results chapter to justify why you are looking more carefully to certain processes

Answer: Line 146: According with reviewer comment the reference was added.

Line 45: C. albicans IS an opportunistic pathogen and becomes pathogenic (rather than harmless commensal) in the conditions described. Remove “everything”.

Answer: According with reviewer comment the change was made.

Line 67-73: Beware that antifungal tolerance and resistance are not the same thing.

Answer: According with reviewer comment the paragraph was adjusted

Line 113-114: “promote the aggregation of cytoplasmic” ??

Answer: According with reviewer comment the paragraph was adjusted

Results:

I found the order in which the results are presented a little confusing. First, you analyse the susceptibility of several mutants to the peptide, extract and fluconazole, then you do proteomics without mentioning these proteins, and finally you go back to them to do more specific analysis on ROS and mitochondrial membrane potential. Maybe the proteomics and more general analysis should come first.

Answer: The experimental approach was thus proposed, taking into account the availability of information regarding the mechanism of action for LfcinB and derived peptides, especially the peptide LfcinB15 (Chang et al., 2021), where similar mutant strains were used and biological assays. For this reason, it was decided to have preliminary information with the use of mutant strains and proteomics to then verify with biological assays, the regulation of the activity of some proteins after incubation with peptide, extract and combination.

Reference: Chang, C.-K., Kao, M.-C., & Lan, C.-Y. (2021). Antimicrobial Activity of the Peptide LfcinB15 against Candida albicans. Journal of Fungi, 7(7), 519. https://doi.org/10.3390/jof7070519

Regarding the mutants:

- the CDR1/CDR2 proteins were named in the proteomics section in table 2, yellow section (transporters) and the ABC transporters are also mentioned in the explanation of the table (lines 336 to 340).

- In the case of MAPKs, Hog1 is mentioned in table 2, also in line 343.

- Finally, proteins involved in electron chain transport are mentioned in general terms, also in table 2, and their description.

Although in some cases, in the proteomic approach, no regulation of the proteins studied in the mutant strains was found, regulation of processes that involve them was observed, and even associated proteins, as seen in table 2.

2.1:  As referred in the Introduction chapter (Line 70-73), resistance to fluconazole can be caused by several mechanisms. How do the authors know that Ca256 strain is resistant to fluconazole due to a mutation on ERG11 (line 143)? Please provide a reference.

Answer: The strain has been sequenced in its entire genome; however, we have had technical problems due to haploidy and at the moment we have not been able to evaluate the ERG11 gene, therefore this information was eliminated from the article.

 Is the info provided in table 1 a review of previously published data [13], which info is? Why there is no data on MIC and MFC (Table 1) for cells treated with both R-1-R and extract (antifungal combination)?

Answer: Only MIC and MFC values were reported in reference 13, this information was added in the description below the table.

The approach with mutant strains focuses on the changes in susceptibility of said mutant strains to a new antifungal molecule, compared to wild type strains. Taking this into account, it was decided to determine only the MICs for peptide and extract individually, where the analysis can be performed with an increase or decrease in the MIC. However, to evaluate the combination, the methodology to be used is different, since the checkerboard methodology is required, where the interpretation is carried out with an index to determine the synergy, additivity, indifference and antagonism, which does not ideally fit what has been proposed for the assay of mutant strains and the interpretation could not be analyzed in the correct way.

However, we performed an assay measuring the change in the susceptibility of the homozygous mutants after treatment with the combination, measuring changes in absrobancies. The results were added: Figure 1, description in lines 192-195 and 207-209.

Please explain/discuss better why do you use heterozygous instead of homozygous mutants and the observed differences in MIC/MFC. Why weren´t CDR1/2 homozygous mutants also used?

Answer: Both heterozygous and homozygous mutant strains were used, but this was not the case for CDR1 and CDR2, since these are not available in the collection of homozygous mutant strains.

The difference in MIC and MFC between homozygous and heterozygous mutant strains may be due to the fact that the allele that is conserved may be supplying the function of the allele that is absent, therefore, the change in susceptibility may be greater or lesser depending on the type of mutant used.

Why were the MAPKs heterozygous mutants more resistant to FLC than the wild type? How do the null mutants behave?

Answer: To further analyze the mutants' susceptibility to stress conditions, we examined their growth inhibition in the presence of fluconazole, which affect the fungal cell membrane either directly or indirectly. Here, our data support previous studies that demonstrate HOG1 deletion was found to be sensitive to polyene class, such as amphotericin B, but no to azole drugs, such as fluconazole, while the deletion of MKC1 did not affect the susceptibility to both antifungals (Correia et al., 2020). Since, The HOG pathway controls expression of ergosterol biosynthesis genes, including ERG11, inhibition of the HOG pathway increases ergosterol, enhancing amphotericin B's effectiveness. Conversely, higher expression of ERG11 necessitates greater concentrations of azole drugs to achieve inhibition, thus reducing their antifungal efficacy (Ko YJ et al., 2009).

Additionally, MAPKs heterozygous mutants were found to be less sensitive to fluconazole (FLC), reinforcing the idea that Hog1 and Mkc1 are key kinases for adapting to resistance against compounds that interact with cell wall integrity or disrupt ergosterol biosynthesis differently from the wild type (Monge et al., 1999). Then, heterozygous mutants lacking one copy of each gene (hog1∆/HOG1 and mkc1∆/MKC1) exhibit changes in membrane permeability. Azole resistance can arise from modifications to the drug target, such as changes in Erg11 expression levels that reduce drug accumulation, alterations in the structure or concentration of antifungal target proteins, or modifications in membrane sterol composition (Sanglard et al., 2002).

References:

  • Correia, I.; Wilson, D.; Hube, B.; Pla, J. Characterization of a Candida albicansMutant Defective in All MAPKs Highlights the Major Role of Hog1 in the MAPK Signaling Network.  Fungi 20206, 230. https://doi.org/10.3390/jof6040230
  • Ko YJ, Yu YM, Kim GB, Lee GW, Maeng PJ, et al. (2009) Remodeling of global transcription patterns of Cryptococcus neoformans genes mediated by the stress-activated HOG signaling pathways. Eukaryot Cell 8: 1197–1217.
  • Alonso-Monge R, Navarro-García F, Molero G, Diez-Orejas R, Gustin M, Pla J, Sánchez M, Nombela C. Role of the mitogen-activated protein kinase Hog1p in morphogenesis and virulence of Candida albicans. J Bacteriol. 1999 May;181(10):3058-68. doi: 10.1128/JB.181.10.3058-3068.1999. PMID: 10322006; PMCID: PMC93760.
  • Sanglard, D., & Odds, F. C. (2002). Resistance of Candida species to antifungal agents: molecular mechanisms and clinical consequences. The Lancet Infectious Diseases, 2(2), 73–85. doi:10.1016/s1473-3099(02)00181-0

Please explain better the hypothesis. This was the hypothesis for the results in paper [13], correct? But how does this explain the increased tolerance of mutants to the extract if FLC is not being simultaneously added?

The hypothesis arises from the results of reference 13 and the information reported by other authors regarding the mechanism of action of LfcinB and peptides derived from LfcinB.

We infer this hypothesis, which could be related to the change in susceptibility to the peptide, and perhaps could be extrapolated to the extract; however, there is no information on the mechanism of antifungal action of B. pilosa extracts, so a clear conclusion regarding what the extract is generating in the efflux pumps cannot yet be obtained without further studies.

Now, in the use of mutant strains as a tool to study the mechanism of action, we consider that we should first try to clarify the role of the extract, and then proceed to add other variables, such as the combination with FLC, for example.

Line 177-179: “…which suggests… decreases.” please remove the sentence, it is unnecessary.

Answer: According with reviewer comment the change was made.

Line 195: consider replacing “novel opportunity” for “alternative method” or something similar. The reference is from 2007.

Answer: According with reviewer comment the change was made.

Have you checked the role of Cek1 MAPK in this response?

Answer: Considering that the research carried out for the peptide LfcinB15 was a point of reference for our work and that previously the activation of Cek1 MAPK was not evidenced for the peptide LfcinB15, in the present work the role of cek1 was not evaluated.

Reference: Chang, C.-K., Kao, M.-C., & Lan, C.-Y. (2021). Antimicrobial Activity of the Peptide LfcinB15 against Candida albicans. Journal of Fungi, 7(7), 519. https://doi.org/10.3390/jof7070519

Table S1: replace CMI by MIC

Answer: According with reviewer comment the change was made.

Figure S1: legend for c) and d)

Answer: According with reviewer comment the information was added.

Lines 227-236: improve. Have you done growth curves or measured the OD? It seems that the observed differences may be due to different growth rates. The drugs inhibit growth, thus less OD after 6h treatment, thus less protein. Are FLC resistant strains more fit? Please explain better how you obtained the samples for proteomic analysis (see Materials and Methods).

Answer: Time kill curves were previously performed for the strains treated with peptide and extract, individually (Vargas-Casanova et al., 2020, 2023), and no significant changes in absorbance were observed before 10 hours of incubation. For this reason, and in order to demonstrate the effects exerted by peptide and extract without the OD affecting the results, the incubation time of the yeasts with peptide and extract in the proteomic assay was 6 hours, as mentioned in item 4.5, line 796.

References:

  • Vargas-Casanova, Y., Bravo-Chaucanés, C. P., Martínez, A. X. H., Costa, G. M., Contreras-Herrera, J. L., Medina, R. F., Rivera-Monroy, Z. J., García-Castañeda, J. E., & Parra-Giraldo, C. M. (2023). Combining the Peptide RWQWRWQWR and an Ethanolic Extract of Bidens pilosa Enhances the Activity against Sensitive and Resistant Candida albicans and C. auris Strains. Journal of Fungi, 9(8), 817. https://doi.org/10.3390/jof9080817
  • Vargas-Casanova, Y., Carlos Villamil Poveda, J., Jenny Rivera-Monroy, Z., Ceballos Garzón, A., Fierro-Medina, R., Le Pape, P., Eduardo García-Castañeda, J., & Marcela Parra Giraldo, C. (2020). Palindromic Peptide LfcinB (21-25)Pal Exhibited Antifungal Activity against Multidrug-Resistant Candida. ChemistrySelect, 5(24), 7236–7242. https://doi.org/10.1002/slct.202001329

2.3: When comparing SC5314 and Ca256 there are more proteins identified in the resistant strain. Either common or exclusive…

Answer: More proteins were identified in the resistant strain, in a basal manner, both in the total proteins (SC5314: 2186 proteins vs 256: 2367) and in the exclusive ones (SC5314: 75 proteins vs 256: 115) (Figure 1)

Tables A from Appendix A.

Line 246-248: either the down-regulation and up-regulation of proteins could compensate for the damage... Please discuss this better. Why is the number of up-regulated and down-regulated proteins important? Did you analyse which specific proteins are having different behaviours?

Answer: The importance of the amount of up- and down-regulated proteins is important since it could be inferred that with up-regulated proteins, it is possible that the yeast is trying to compensate for the biological processes that are being affected by the treatment, on the contrary, with down-regulated proteins, it could be suggested that the damage caused by the treatment is so significant that the yeast cannot recover from this effect.

Taking into account the above, the conclusions suggested are made in:

  • Line 245-247 are made: .... “both strains show a higher number of up-regulated proteins compared to down-regulated proteins, the fact that more up-regulated proteins were found when the strains were incubated with R-1-R suggests that the yeast could be generating a compensatory response to the damage caused by the peptide….”
  • Line 384-386: ….” This suggests that the combination treatment suppresses a greater number of proteins, potentially indicating a reduced ability of strain SC5314 to mount a compensatory re-sponse.”…….

Additionally, specific proteins with different behaviors (up and down regulated) are summarized in Tables 2 and 3, and in Figures 11 and 12.

Where did you get the information on column 6 “relationship with antifungal activity”?

Answer: The information on column 6, is the categorization carried out, according to the biological functions of the proteins, which arose from the information on column 5. This categorization was also carried out by relating the function of the proteins with the information reported regarding the mechanism of action for LfcinB and derived peptides.

Why don´t you show “others” as separate categories since these are the most represented?

Answer: This category was not separated, since many biological processes included here tend to be very general or not specific associated with any particular organelle or biological pathway, however, in the future it is considered to analyze in a more exhaustive way the proteins involved included in this category.

Please revise the tables. For instance, Mitochondria – ox stress (purple) states as 13 entries but the purple cells also contain “others” and “unknown”.

Answer: According with reviewer comment the change was made.

2.4:  Which specific proteins change their expression when the extract is added (up-regulated with the peptide but down-regulated with the antifungal combination? I think it would be more interesting if comparisons were made directly between SC5314 + R-1-R and SC5314 + R-1-R + extract.

Answer: In Item 2.6 . it is described not only the important proteins up- or down-regulated for SC5314 in response to the combination, but also the differences in regulation when SC5314 is treated with peptide alone.

2.5: Why is the number of total SC5314 basal proteins in figure 4 different from figure 1?

Answer: This occurs because although the total number of quantified proteins is the same in SC5314 (2424 proteins), the basal protein value is obtained by subtracting the total number of quantified proteins (2424) from the exclusive proteins regulated by the peptide (238) or the combination (115). It should be noted that the values for exclusive proteins are obtained after the label-free quantitative analysis. Thus, when comparing basal SC5314 with the peptide treatment, a total of 2186 (2424-238) basal proteins were identified (figure 1), and when comparing basal SC5314 with the combination, 2309 (2424-115) total basal proteins were identified (figure 4).

2.6: Consider showing the list of all exclusively regulated proteins by the antifungal combination. How was the choice of that small subset of proteins made?

The table 3 include all exclusively regulated proteins by the antifungal combination in SC5314. In the lines 433-435, the explanation about how the proteins were chosen was described:

“Important proteins that were exclusively regulated by the combination treatment and absent when SC5314 was treated with R-1-R alone were selected for further analysis. A smaller subset of proteins was chosen (Table 3)”.

2.7: Please discuss better the results shown in Figure 8, particularly those seen for the extract treatment.

Answer: According with reviewer comment the discussion was improved. Lines 539-550

Shouldn’t the values be normalized?

Answer: We considered that this experiment may not require normalization for several reasons: 1. The experimental conditions are controlled and consistent across all samples and time points, ensuring that observed changes are due to the experimental variable, not setup variations. 2. Our goal is to compare raw data directly between two samples, where normalization might obscure important differences. Absolute concentration values over time can provide more insight in kinetic studies. 3. Raw data might be more informative and reliable for specific analyses. Normalization, involving mathematical manipulation, can introduce errors or misinterpretations if not done carefully. 4. If baseline measurements (e.g., initial concentrations) are stable and consistent across samples, the need for normalization is reduced. Also, normalization can add unnecessary complexity, especially in early research stages. Finally, we noticed than over 95% of research involving R6G has been conducted without normalizing the data. This approach is supported by the fact that raw fluorescence measurements are generally adequate for effects of R6G across various studies. This established practice in research supports the continued use of raw data to maintain consistency with established methodologies.

2.8:  Have you checked the phosphorylation state of the CaMAPKs in the presence of the peptide, extract and both?

Answer: Phosphorylation experiments have not been carried out yet, however, it is one of the perspectives of the research and we hope to do it in the future.

Materials and Methods:

How was GO enrichment performed? Which software and which background (all Ca entries or only those proteins identified by MS?).

Answer: We consider that "Go enrichment analysis" is sufficiently explained in the manuscript in the section "4.8 Bioinformatics analysis", taking as a reference point the majority of papers related to this type of study.

The software used is the gprofiler2 package (Line 862)

This analysis was carried out taking into account the set of proteins that are "up regulated" and "down regulated" in each of the conditions (this is described in lines 858-860), obtaining the terms for which our set of proteins are over- or under-represented. The technical basis behind this explanation is a hypergeometric test, the result of which was the p value associated with each term, which was later taken into account to compile the list of biological terms (this is summarized in line 865).

4.5: please explain better. Pre-inoculum in what, for how long at which temperature? Dilution in RPMI plus drug (or not) and incubated for 6h?

Answer: According with reviewer comment the information was added.

Final OD measurements to correlate with protein concentration?

Answer: The proteomic approach was not performed with an OD measurement, the iniculus was prepared and quantified by counting the number of cells in a Neubauer chamber, this is described in the materials and methods section.

However, according to the growth curves previously performed, it was evident that there is no significant variation in the OD until approximately 10 hours of incubation, between treated and untreated cells.

4.6: How did you obtain the peptides? Enzyme used, in-gel or in-solution digestion?

Answer: The peptides were obtained by digestion and desalting with the PreOmics®' iST Kit, as mentioned in lines 813 and 814.

4.8: How did you determined overrepresentation?

 Answer: In section 4.7., the procedure for protein expression analysis is explained, using Quantification by Label Free and in lines 850 to 854, it is explained how the differential proteins (Up and Down-regulated proteins) were determined: “The filters applied to obtain the list of differential proteins (Up and Down-regulated proteins) were: (i) Abundance ratio variability: Variability of the ratios calculated as a percentage must present a value below 30% and above 0%. This last case ensured elimination of those proteins that have been quantified from a single measurement. (ii) p-value <0.05, and (iii) q-value <0.05 [64].”

Other comments:

Data availability: is the proteomic data available at PRIDE or are you planning to submit it to a public repository?

 Answer: According with reviewer comment the information was added in the sections: abstract (Line 30) and Data Availability Statement.

Comments on the Quality of English Language and Formatting

I found the manuscript confusing and difficult to follow primarily due to que quality of the English. It would benefit from a thorough language revision to improve its clarity and readability.

Answer: According with reviewer comment a detailed review of the entire document was carried out again, as well as multiple adjustments that allow for a clear understanding in terms of the English language.

Line 3: “Bidens pilosa” in italic. Please review this in the entire manuscript including references.

Answer: According with reviewer comment the change was made.

Line 5-8: check format

Answer: According with reviewer comment the format was cheked.

Line 21-22: “studied” … ”study”… “study”..

Answer: According with reviewer comment the changes were made.

Line 48: remove “causing”

Answer: According with reviewer comment the change was made.

Line 49: remove ";”

Answer: According with reviewer comment the change was made.

Line 131: “…a metabolite present in a…”

Answer: According with reviewer comment the change was made

Line 209 (Table 1): Consider changing orf19 id for recent systematic name (Assembly 22); Please replace “CMI/CMF” for “MIC/MFC”; Oral clinical isolate, not “isolated”; …was obtained FROM [31]; Not determined.

Answer: According with reviewer comment the changes were made.

Line 210-211: Format and remove the dot at proteins.

Answer: According with reviewer comment the changes were made.

Line 759: according TO the protocol

Answer: According with reviewer comment the word was added

Line 769: are the references correct?

Answer: According with reviewer comment, the references were reviewed and are correct.

Line 771: SDA meaning

Answer: According with reviewer comment, the meaning was added.

Line 871: SDA?

Answer: According with reviewer comment, the meaning was added in the Line 777

Comments on the Quality of English Language

I found the manuscript confusing and difficult to follow primarily due to que quality of the English. It would benefit from a thorough language revision to improve its clarity and readability.

Answer: According with reviewer comment a detailed review of the entire document was carried out again, as well as multiple adjustments that allow for a clear understanding in terms of the English language.

Reviewer 2 Report

Comments and Suggestions for Authors

Vargas-Casanova et al manuscript describes the use of proteomic analysis and of other assays to confirm the antifungal mode of action of the 34 R-1-R peptide in Candida albicans. Materials and methods contain the required detail to reproduce the work, results are clearly presented and described. 

I suggest the following revisions:

-       Line 3 (title): Bidens pilosa should be in italics

-       The MS has two sections with discussions “2. Results and discussion” and “3. Discussion” and another section of “4. Conclusion”. The MS is too long.

-       Table 1: it is written CMI and CMF instead of MIC and MFC.

-       Line 210: the 2.2 section title is in the caption of the table

-       Line 211: protei.ns

-       Line 212-216: Please rephrase

-       Line 318: please introduce which software was used

-       Line 334-349: please rephrase

-       Line 361-364: is it a caption from the tables?

-       Section 2.7 Scanning transmission electron microscopy? Is it scanning or transmission electron microscopy? By the figures 6 and 7, it seems to me transmission electron microscopy but contradictory, in the methodology the microscope describe is a scanning electron microscope. 

-       Line 468: please indicate in the figure the cytoplasmic microbodies with arrows

-       Figure 6 and 7: no scale bars are visible

-       Line 530: AmB first citation (explanation is line 901)

-       Line 522 and 649: What PAMs stands for?

-       In some places it is written B. Pilosa instead of B. pilosa

-       Figure 11: “citoesqueleto e integridad”, please translate

-       Line 777: RPMI first citation (line 788) but Roswell Park Memorial Institute is line 788

-       Section 4.9: the samples treatment after glutaraldehyde step and PBS washing is not described

-       Section 4.10 is described as a lab protocol

In all the MS:

-       It is imperative that the entire article is revised by a native English speaker. Examples: line 128: “To extracts,”, line 135 “the changes in protein expression alteration”, line 301 “140 proteins were involved”, line 330 “Erg11p did not appear”, line 341 “oxidoreductase activity proteins were down”, line 376: “the regulation of proteins was opposite to the results obtained with the peptide R-1-R, where more proteins had been obtained Up-regulated than Down-regulated”, line 649: “is directed at the cell surface”, line 736 “the proteomic approach revealed that parallel to the Hog1 pathway”

-       Why using capital letters for Down and Up regulated proteins in some places?

Comments on the Quality of English Language

 It is imperative that the entire article is revised by a native English speaker. There are some examples: line 128: “To extracts,”, line 135 “the changes in protein expression alteration”, line 301 “140 proteins were involved”, line 330 “Erg11p did not appear”, line 341 “oxidoreductase activity proteins were down”, line 376: “the regulation of proteins was opposite to the results obtained with the peptide R-1-R, where more proteins had been obtained Up-regulated than Down-regulated”, line 649: “is directed at the cell surface”, line 736 “the proteomic approach revealed that parallel to the Hog1 pathway”

Author Response

Response to Reviewer 2 Comments

Comments and Suggestions for Authors

Vargas-Casanova et al manuscript describes the use of proteomic analysis and of other assays to confirm the antifungal mode of action of the 34 R-1-R peptide in Candida albicans. Materials and methods contain the required detail to reproduce the work, results are clearly presented and described. 

I suggest the following revisions:

-       Line 3 (title): Bidens pilosa should be in italics

Answer: According with reviewer comment the change was made.

-       The MS has two sections with discussions “2. Results and discussion” and “3. Discussion” and another section of “4. Conclusion”. The MS is too long.

Answer: The document contains a lot of information that we consider important, we have decided to organize the paper in this way with the guide of other article previously published by other authors in the same journal (Kolić & Šinko, 2024) and that present a similar thematic content regarding the biological activity of another compound. Section 2. Results and discussion, describes the values and results obtained and also makes the comparison with previous studies; in section 3. Discussion, we added the word "General" to the title, in this section we have tried to unify and correlate the results with each other, in order to synthesize and finally conclude in section 4. regarding the new knowledge obtained in this research.

Reference: Kolić, D., & Šinko, G. (2024). Evaluation of Anticholinesterase Activity of the Fungicides Mefentrifluconazole and Pyraclostrobin. International Journal of Molecular Sciences, 25(12), Article 12. https://doi.org/10.3390/ijms25126310

-       Table 1: it is written CMI and CMF instead of MIC and MFC.

Answer: According with reviewer comment the changes were made.

-       Line 210: the 2.2 section title is in the caption of the table

Answer: According with reviewer comment the change was made.

-       Line 211: proteins

Answer: According with reviewer comment the change was made.

-       Line 212-216: Please rephrase

Answer: According with reviewer comment the change was made.

-       Line 318: please introduce which software was used

Answer: The information is mentioned in section 4.8 Bioinformatics analysis: …. . “To predict the protein-protein interaction networks, the STRING database (https://string-db.org/) was used”….

Additionally, .... "The STRING" was added to figures 3 and 5

-       Line 334-349: please rephrase

Answer: According with reviewer comment the change was made.

-       Line 361-364: is it a caption from the tables?

Answer: The paragraph corresponds to a description of the table, the format was adjusted.

-       Section 2.7 Scanning transmission electron microscopy? Is it scanning or transmission electron microscopy? By the figures 6 and 7, it seems to me transmission electron microscopy but contradictory, in the methodology the microscope describe is a scanning electron microscope. 

Answer: “Scanning transmission electron microscopy (STEM) is a combination of SEM and TEM: that is, a transmission image is obtained using a scanning method”. (X Zhou, G.E. Thompson, in Reference Module in Materials Science and Materials Engineering, 2017)

In the following link you can find several STEM references: https://www.sciencedirect.com/topics/materials-science/scanning-transmission-electron-microscopy

-       Line 468: please indicate in the figure the cytoplasmic microbodies with arrows

Answer: According with reviewer comment arrows were added.

-       Figure 6 and 7: no scale bars are visible

Answer: According with reviewer comment scale bars were added.

-       Line 530: AmB first citation (explanation is line 901)

Answer: According with reviewer comment the change was made.

-       Line 522 and 649: What PAMs stands for?

The correct abbreviation is: AMPs : antimicrobial peptides

Answer: According with reviewer comment the change was made.

-       In some places it is written B. Pilosa instead of B. pilosa

Answer: According with reviewer comment the changes were made.

-       Figure 11: “citoesqueleto e integridad”, please translate

Answer: According with reviewer comment the change was made.

-       Line 777: RPMI first citation (line 788) but Roswell Park Memorial Institute is line 788

Answer: According with reviewer comment the change was made.

-       Section 4.9: the samples treatment after glutaraldehyde step and PBS washing is not described

Answer: According with reviewer comment the information was added.

Section 4.9.: …. “The pellets were fixed, desiccated, and embedded, as previously described [65]. Then the cells were observed”…

-       Section 4.10 is described as a lab protocol

Answer: According with reviewer comment the description was modified.

In all the MS:

-       It is imperative that the entire article is revised by a native English speaker. Examples: line 128: “To extracts,”, line 135 “the changes in protein expression alteration”, line 301 “140 proteins were involved”, line 330 “Erg11p did not appear”, line 341 “oxidoreductase activity proteins were down”, line 376: “the regulation of proteins was opposite to the results obtained with the peptide R-1-R, where more proteins had been obtained Up-regulated than Down-regulated”, line 649: “is directed at the cell surface”, line 736 “the proteomic approach revealed that parallel to the Hog1 pathway”

 Answer: According with reviewer comment the change were made.

-       Why using capital letters for Down and Up regulated proteins in some places?

Answer: According with reviewer comment the change were made.

Comments on the Quality of English Language

 It is imperative that the entire article is revised by a native English speaker. There are some examples: line 128: “To extracts,”, line 135 “the changes in protein expression alteration”, line 301 “140 proteins were involved”, line 330 “Erg11p did not appear”, line 341 “oxidoreductase activity proteins were down”, line 376: “the regulation of proteins was opposite to the results obtained with the peptide R-1-R, where more proteins had been obtained Up-regulated than Down-regulated”, line 649: “is directed at the cell surface”, line 736 “the proteomic approach revealed that parallel to the Hog1 pathway”

Answer: According with reviewer comment the change were made.
